# ImageNet−X: Understanding Model Mistakes with Factor of Variation Annotations

**Badr Youbi Idrissi, Diane Bouchacourt, Randall Balestriero, Ivan Evtimov, Caner Hazirbas
Nicolas Ballas, Pascal Vincent, Michal Drozdzal, David Lopez-Paz, Mark Ibrahim**[*]
Fundamental AI Research (FAIR), Meta AI
{byoubi,marksibrahim}@meta.com

## Abstract

Deep learning vision systems are widely deployed across applications where reliability is critical. However, even today's best models can fail to recognize an object when its pose, lighting, or background varies. While existing benchmarks surface examples that are challenging for models, they do not explain why such mistakes arise. To address this need, we introduce `ImageNet-X`–a set of sixteen *human* annotations of factors such as pose, background, or lighting for the entire ImageNet-1k validation set as well as a random subset of 12k training images. Equipped with `ImageNet-X`, we investigate 2,200 current recognition models and study the types of mistakes as a function of model's (1) architecture – e.g. transformer vs. convolutional –, (2) learning paradigm – e.g. supervised vs. self-supervised –, and (3) training procedures – e.g. data augmentation. Regardless of these choices, we find models have consistent failure modes across `ImageNet-X` categories. We also find that while data augmentation can improve robustness to certain factors, they induce spill-over effects to other factors. For example, color-jitter augmentation improves robustness to color and brightness, but surprisingly hurts robustness to pose. Together, these insights suggests that to advance the robustness of modern vision models, future research should focus on collecting additional diverse data and understanding data augmentation schemes. Along with these insights, we release a toolkit based on `ImageNet-X` to spur further study into the mistakes the image recognition systems make: `https://facebookresearch.github.io/imagenetx/site/home`.

## 1 Introduction

Despite deep learning surpassing human performance on ImageNet (Russakovsky et al., 2015; He et al., 2015), even today's best vision systems can fail in spectacular ways. Models are brittle to variation in object pose (Alcorn et al., 2019), background (Beery et al., 2018), texture (Geirhos et al., 2018), and lighting (Michaelis et al., 2019).

Model failures are of increasing importance as deep learning is deployed in critical systems spanning fields across medical imaging (Lundervold and Lundervold, 2019), autonomous driving (Grigorescu et al., 2020), and satellite imagery (Zhu et al., 2017). One example from the medical domain raises reasonable worry, as "recent deep learning systems to detect COVID-19 rely on confounding factors rather than medical pathology, creating an alarming situation in which the systems appear accurate, but fail when tested in new hospitals" (DeGrave et al., 2021). Just as worrisome is evidence that model failures are pronounced for socially disadvantaged groups (Chasalow and Levy, 2021; Buolamwini and Gebru, 2018; DeVries et al., 2019; Idrissi et al., 2021).

Existing benchmarks such as ImageNet-A,-O, and -V2 surface more challenging classification examples, but do not reveal why models make such mistakes. Benchmarks don't indicate whether a model's failure is due to an unusual pose or an unseen color or dark lighting conditions. Researchers, instead, often measure robustness with respect to these examples' average accuracy. Average accuracy captures a model's mistakes, but does not reveal directions to reduce those mistakes. A hurdle to research progress is understanding not just *that*, but also *why* model failures occur.

---

[*]hands-on and advising contributions

To meet this need, we introduce `ImageNet-X`, a set of *human* annotations pinpointing failure types for the popular ImageNet dataset. `ImageNet-X` labels distinguishing object factors such as pose, size, color, lighting, occlusions, co-occurences, and so on for each image in the validation set and a random subset of 12,000 training samples. Along with explaining how images in ImageNet vary, these annotations surface factors associated with models' mistakes (depicted in Figure 1).

a) `ImageNet-X` annotation form

Prototypes            Sample

Given the group of `Prototypical` images to the left and `Sample` image to the right, please answer the following questions in blue:

1. Select all factors that make the `Prototypical` images different from the `Sample` image: ☑ pose / positioning; ☑ object is partially present; ☑ object partially blocked by another object; ☐ object partially blocked by a person; ☐ another object is present; ☑ object is small relative to the image frame; ☐ object is large relative to the image frame; ☐ lightning is brighter; ☐ lightning is darker; ☑ background; ☑ color; ☐ shape; ☐ texture; ☐ pattern; ☐ media style; ☐ subcategory.

2. Describe more about your selections in question 2: cow in water at the beach

3. Describe in one word what the primary difference is between the left images and right image: beach, far-away

b) **Robustness analysis enabled by `ImageNet-X`**

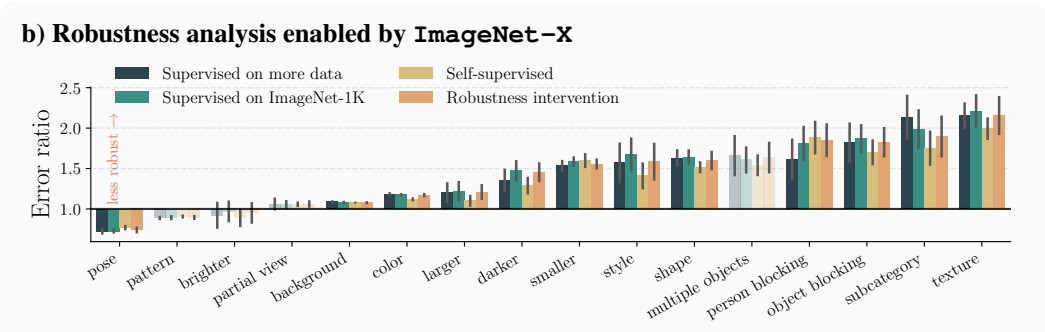

Figure 1: **Models, regardless of architecture, training dataset size, and even robustness interventions all share similar failure types.** `ImageNet-X` annotations allow us to group images into *Factors of Variation* such as pose, pattern or texture (subfigure **a** and full definitions in Appendix A.2). A model can be evaluated on each of these factors, revealing where it makes the most mistakes. We compare error ratios $= \frac{1-\mathrm{acc}(factor)}{1-\mathrm{acc}(overall)}$ on each factor for 4 wide groups of models. Subfigure **b** shows that differences in texture, subcategories (e.g., breeds), and occlusion are most associated with models' mistakes. Transparent bars show the factors where there is no significant difference between the 4 groups (p value $> 0.05$ with Alexander Govern test).

By analyzing the `ImageNet-X` labels, in section 3, we find that in ImageNet pose and background commonly vary, that classes can have distinct factors (such as dogs more often varying in pose compared to other classes), and that ImageNet's training and validation sets share similar distributions of factors. We then analyze, in section 4, the failure types of more than 2,200 models. We find that models, regardless of architecture, training dataset size, and even robustness interventions all share similar failure types in section 4.1. Additionally, differences in texture, subcategories (e.g., breeds),

| Dataset | Description | ImageNet images | Natural images | Entire val. set | Human FoV | Ref. |
|---|---|:---:|:---:|:---:|:---:|---:|
| ImageNet-C | algorithmic corruptions | ✗ | ✗ | ✓ | ✗ | Hendrycks and Dietterich (2019) |
| ImageNet-P | animated perturbations | ✗ | ✗ | ✓ | ✗ | Hendrycks and Dietterich (2019) |
| ImageNet-R | artistic renditions | ✗ | ✗ | ✗ | ✗ | Hendrycks et al. (2021) |
| ImageNet-A | natural adversarial examples | ✗ | ✓ | ✗ | ✗ | Hendrycks et al. (2021) |
| ImageNet-O | out-of-distribution examples | ✗ | ✓ | ✗ | ✗ | Hendrycks et al. (2021) |
| ImageNet-V2 | new validation set | ✗ | ✓ | ✗ | ✗ | Recht et al. (2019) |
| ImageNet-Sketch | drawn sketches | ✗ | ✗ | ✗ | ✗ | Wang et al. (2019) |
| ImageNet-ML | human multi-labels | ✗ | ✓ | ✗ | ✗ | Shankar et al. (2020) |
| ImageNet-ReaL | human multi-labels | ✓ | ✓ | ✓ | ✗ | Beyer et al. (2020a) |
| ImageNet-ReLabel | machine pixelwise multilabels | ✓ | ✓ | ✓ | ✗ | Yun et al. (2021) |
| ImageNet-Stylized | randomly-textured images | ✓ | ✗ | ✓ | ✗ | Geirhos et al. (2018) |
| ImageNet-X | human FoV annotations | ✓ | ✓ | ✓ | ✓ | ours |

Table 1: Extensions of the ImageNet benchmark extensions designed to inspect failure modes of the ImageNet trained models. We characterize each dataset by looking whether : (1) the dataset images are only coming from the ImageNet validation set — **ImageNet images** —; (2) the dataset images are natural images or are created with algorithmic and artistic perturbations of natural images — **Natural images** —; (3) the dataset annotates the entire ImageNet validation set — **Entire val. set** —; and (4) whether the dataset contains human annotations of image factors of variations — **Human FoV** —. Our proposed ImageNet-X is the first dataset based on ImageNet to include human annotations of multiple factors of variation for the entire ImageNet validation set.

and occlusion are most associated with models' mistakes (See Figure 1 and section 4.3.1). Among modeling choices such as architecture, supervision, data augmentations, and regularization methods, we find data augmentations can boost models' robustness. Common augmentations such as cropping and color-jittering however, can have unintended consequences by affecting unrelated factors (see section 4.3.2). For example, cropping improves robustness to pose and partial views, as expected, all the while affecting unrelated factors such as pattern, background, and texture. Together these findings suggests that to advance the robustness of modern vision models, future research should focus on improving training data – by collecting additional data and improving data augmentation schemes – and deemphasize the importance of other aspects such as choice of architecture and learning paradigm.

We release all the ImageNet-X annotations along with an open-source toolkit to probe existing or new models' failure types. The data and code are available at

https://facebookresearch.github.io/imagenetx/site/home.

With ImageNet-X we equip the research community with a tool to pin-point a models' failure types. We hope this spurs new research directions to improve the reliability of deep learning vision systems.

## 2 RELATED WORK

The research community has developed approaches for testing models' robustness on extended versions of the ImageNet dataset – see Figure 3 for a visual on different levels of evaluation granularity and Table 1 for an overview of datasets created to test the failure modes of the models trained on the ImageNet data. One common approach is to introduce artificial augmentations , e.g. image corruptions and perturbations (Hendrycks and Dietterich, 2019), renditions (Hendrycks et al., 2021), sketches (Wang et al., 2019; Bordes et al., 2021), etc. These artificial variations capture changes arising from corruptions, but are unlikely to capture changing arising from natural distribution shifts or variation such as changes in pose, lighting, scale, background etc. Consequently, researchers also collected additional natural images to study the performance of the classification models under moderate to drastic distribution shifts Hendrycks and Dietterich (2019); Recht et al. (2019). However, most of these datasets are built to assess the model performance when going away from training data distribution and, thus, provide almost no understanding about the nature of the *in-distribution* errors. Currently, the only ImageNet extensions that help analyzing the in-distribution model errors are the multiclass relabelling or saliency of the validation set (Shankar et al., 2020; Beyer et al., 2020a; Yun et al., 2021; Singla and Feizi, 2021). However, this relabelling only explains one type of model error

that is caused by the co-occurrences of other objects in the scene. Our contribution, `ImageNet-X`, builds on this line of work to provide granular labels for naturally occurring factors such as changes in pose, background, lighting, scale, etc. to pinpoint the underlying modes of failure.

## 3 IMAGENET-X: ANNOTATING IMAGENET WITH VARIATION LABELS

`ImageNet-X` contains human annotations for each of the 50,000 images in the validation set of the ImageNet dataset and 12,000 random sample from the training set. Since it's difficult to annotate factors of variations by looking at a single image in isolation, we obtain the annotation by comparing a validation set image to the three class-prototypical images and ask the annotators to describe the image by contrasting it with the prototypical images. We define the prototypical images as the most likely images under ResNet-50 model[1] (He et al., 2015). Trained annotators select among sixteen distinguishing factors, possibly multiple, and write a text description as well as one-word summaries of key differences. The form is illustrated in Figure 1. The factors span pose, various forms of occlusion, styles, and include a subcategory factor capturing whether the image is of a distinct type or breed from the same class (full definitions in Appendix A.2). The text descriptions account for factors outside the sixteen we provide. After training the annotators and verifying quality with multi-review on a subset, each image is annotated by one trained human annotator. For example, the annotator marks whether the object depicted in the `Sample` image is larger or more occluded than `Prototypical` images. We provide a datasheet following Gebru et al. (2021) in Appendix A.1.

**One word summaries confirm the list of factors considered.** Since our pre-selected list of 16 factors may not encompass every type of variation needed to account for model bias, we also asked annotators to provide one-word summaries to best distinguish a given image from its prototypes. We assess whether these free-form responses are encompassed within the pre-defined categories. We find the top-20 one-word annotation summaries are: pattern, close-up, top-view, front-view, grass, black, angle, white, color, background, brown, blue, red, position, facing-left, trees, person, side-view, low-angle, all falling within the 16 categories defined. For example top-view, front-view, facing-left, low-angle, side-view are captured by the *pose* factor.

### 3.1 WHAT IMAGENET-X REVEALS ABOUT IMAGENET

To better understand the proposed annotations, we explore the distribution of the different factors among ImageNet images. We identify the most common varying factors in ImageNet, confirm factor training and validation set distributions match, and find factors can vary in ways that are class-specific.

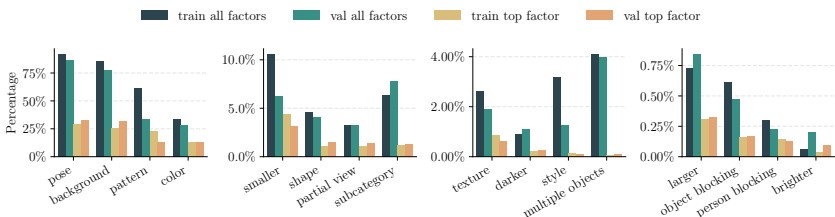

Figure 2: **Some factors are selected for most images; choosing the top factor allows to focus on the main change in the image.** Figure shows the distribution of each factor in the training and validation set both with all factors selected and only top factor selected.

**Pose and background are commonly selected factors.** As we can see in Figure 2, when aggregating all the annotations, pose, background, pattern, and color factors are commonly selected. For instance, pose and background are active for around 80% of the validation images. Since images are not likely to share a background and the objects within are unlikely to share the same pose, annotators selected these factors for many images. Pattern and color, which are the next most common, are also unlikely to be identical across images.

---

[1]The prototypical images for a class are those correctly classifed with the highest predicted probability by ResNet-50. For 115 models with accuracies better than ResNet-50's, the average accuracy on the prototypical examples is 99.4%. Therefore all good models seem to agree on this set of prototypes.

**Selecting the top factor per image.** While the annotators could select multiple factors, we select a unique, *top factor* per image. To do so, we use the free-from text justification that the annotator provided. We embed the text justification using a pretrained language model provided by Spacy (Honnibal and Montani, 2017), and we compare it to that of selected factors' embeddings. This selection allows us to extract the *main* change in each image, thus avoiding over-representing factors that are triggered by small changes, such as pose or background. We see a clear reduction of those factors in Figure 2 (all vs top factors).

**Training and validation sets have similar distributions of factors** To see if there is a distribution shift between training and validation, we annotate a subset of 12k training samples in a similar fashion as the validation[2]. Figure 2 reports the counts of each factor in this subset (denoted *train all* and *train top*). Comparing with the validation dataset, we see most represented factors are very similar to the validation set. We confirm this statistically by performing a $\chi^2$-test on the factor distribution counts, confirming we cannot reject the null hypothesis that the two distributions differ (with $p < 0.01$). Similarly, in Appendix A.3 we see that the distribution of factors ticked by the human annotator (referred to as *active*) per image are close.

To ease our analysis, we consider for each image its meta-label. A meta-label regroups many ImageNet classes into bigger classes. We consider 17 meta-labels that we extract using the hierarchical nature of ImageNet classes: we select 16 common ancestors in the wordnet tree of the original ImageNet 1000 classes. These chosen meta-labels are : device, dog, commodity, bird, structure, covering, wheeled vehicle, food, equipment, insect, vehicle, furniture, primate, vessel, snake, natural object, and other for classes that don't belong to any of the first 16.

**Classes can have distinct variation factors.** In Appendix A.4 (right) we also observed statistically significant correlations between factors and meta-labels. For instance, the dog meta-label is negatively correlated with pattern, color, smaller, shape, and subcategory while being positively correlated with pose and background. This suggests that the images of dogs in the validation set have more variation for pose and background while having less variation for pattern, color, smaller, shape, and subcategory. The commodity meta-label, which contains clothing and home appliances, is positively correlated with pattern and color and negatively correlated with pose.

In addition to revealing rich information about ImageNet, `ImageNet-X` can be used to probe at varying levels of granularity a model's robustness (see Figure 3).

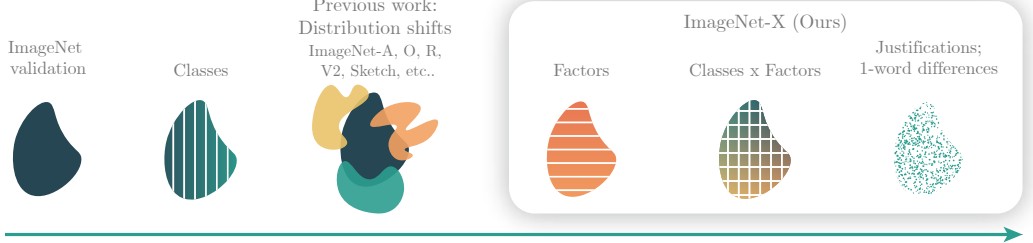

Figure 3: Performance can be evaluated at different levels of granularity. `ImageNet-X` provides factors of variation – aspects that makes an image different from typical images of its class. `ImageNet-X` allows to more precisely pinpoint models' weaknesses and compare them.

## 4 PROBING MODEL ROBUSTNESS

By measuring when factors appear among a model's mistakes relative to the overall dataset, we can characterize any ImageNet model's robustness across the 16 `ImageNet-X` factors. Robustness

---

[2]In the training annotation scheme, we separated "pose" into two factors: "pose" and "subject location", where the latter covers cases where the subject in the same pose but a different part of the image. We reconcile the two here into a unique "pose" to compare with the validation annotations.

via `ImageNet-X` goes beyond revealing model mistakes to pinpointing the underlying factors associated with each mistake.

Here we systematically study the robustness of 2,200 trained models – including many models from the ImageNet test bed Taori et al. (2020a) and additional self-supervised architectures such as DINO and SimCLR – to reveal the limitations of average accuracy, understand model biases, and characterize which choices in architecture, learning paradigm, data augmentation, and regularization affect robustness. For our analysis we focus on the most salient factor for each image by ranking the selected factors by their similarity to the text-justification.

## 4.1 MANY DEEP LEARNING MODELS HAVE THE SAME WEAKNESSES AND STRENGTHS

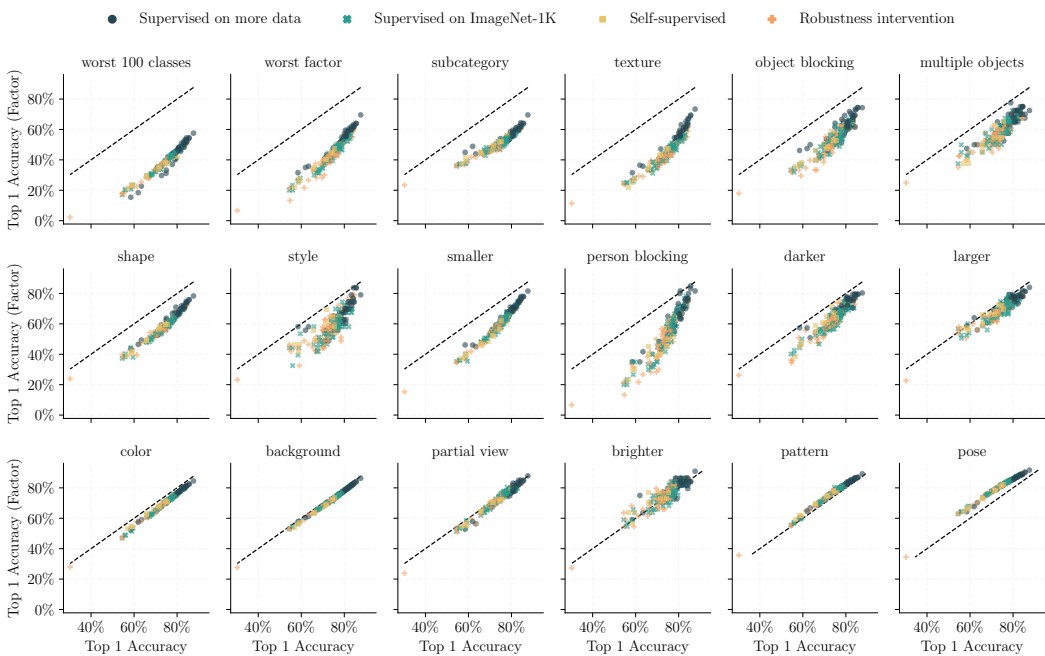

Figure 4: **Most deep learning models, when trained, finetuned or evaluated on ImageNet, have the same biases**. We plot top-1 accuracy for the subset of images labeled with the given factor (y-axis) relative to overall top-1 accuracy (x-axis). The dashed line is an ideal robust models' performance, i.e. performance on each factor is the same as the overall performance. We show the performance of 209 models. We also show the accuracy for the worst factor, and for the images of the worst 100 classes.

To what extent does the commonly reported average accuracy capture a model's robustness? To answer this, we inspect 209 models from the ImageNet testbed Taori et al. (2020a) which includes many architectures (most of which are convolutional, with a few vision transformers), training procedures (losses, optimizers, hyperparameters), and pretraining data. We include additional self supervised models for completeness.

**Models with similar overall accuracies have very similar per factor accuracies.** With all this variety, the scatter plots in Figure 4 exhibit surprising consistency; overall accuracy is a good predictor of per factor accuracies. Most models, even with improving overall accuracy, seem to struggle with the same set of factors: subcategory, texture, object blocking, multiple objects and shape. Conversely, they seem to do well on the same set of factors: pose, pattern, brighter, partial view, background, color, larger, and darker. There are some factors where state-of-the-art models seem to have closed the robustness gap such as person blocking or style.

**More training data helps, but robustness interventions do not.** Models trained with larger datasets (blue circles in Figure 4) exhibit higher accuracy across the factors suggesting larger training datasets do help as others have shown Taori et al. (2020b). Surprisingly, models trained with

robustness interventions (such as CutMix, AdvProp, AutoAugment, etc...), which are directly aimed at improving robustness don't show a significant improvement in per factor accuracy as prior work also shows Taori et al. (2020b).

**Model weaknesses coincide with labeling errors.** The ImageNet labels are known to contain labeling errors Beyer et al. (2020b). With new labels from previous work, we assess the accuracy of the original ImageNet labels and study the distribution of errors across factors. Interestingly, we find that the original label accuracies on all factors coincide with the state of the art models. This offers a potential explanation for the model biases. We leave it to future work to investigate if training on more accurate labels leads to more robust models. We provide details in appendix A.8.

**Is accuracy enough?** Since our annotations are on the ImageNet validation set, a 100% overall accuracy necessarily means a 100% per factor accuracy. So as models get better overall, they necessarily get better per factor accuracies. To disentangle performance from robustness we need to investigate the distribution of errors across factors for a given model.

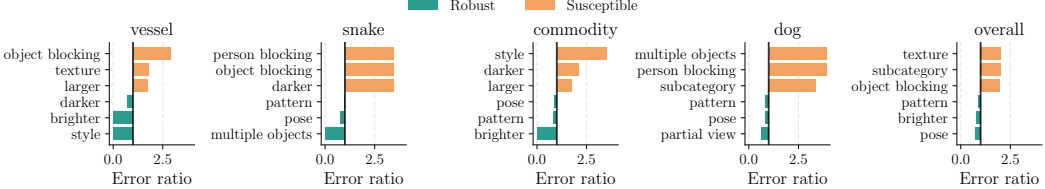

Figure 5: **An illustration of how `ImageNet-X` can identify robustness weaknesses and strengths for Vision Transformers (ViT).** Here we visualize the 3 most susceptible, and 3 most robust factors for the three worst-performing metalabels along with dog. We also include the overall validation set for comparison. We see ViT is susceptible to texture, occlusion, and subcategory, but is robust to pattern, brightness, and pose. While those overall types also show up across metalabels, we see robustness can be distinct by metalabel.

## 4.2 MOVING BEYOND AVERAGE ACCURACY: ASSESSING FAILURE TYPES WITH IMAGENET-X

With `ImageNet-X` we can go beyond average accuracy, to identify the types of mistakes a model makes. To do so we measure the error ratio across each of the 16 `ImageNet-X` factors: Specifically,

$$\text{Error ratio}(\text{factor}, \text{model}) = \frac{1 - \text{accuracy}(\text{factor}, \text{model})}{1 - \text{accuracy}(\text{model})} = \frac{\hat{P}(\text{factor}|\text{correct}(\text{model}) = 0)}{\hat{P}(\text{factor})}.$$

This quantifies how many more mistakes a model makes on a given factor relative to its overall performance. It also measures the increase or decrease in likelihood of a factor when selecting a model's errors vs the overall validation set. A perfectly robust model would have the same error rate across all factors, yielding error ratios of 1 across all factors.

### 4.2.1 AN EXAMPLE OF VISION TRANSFORMER'S MISTAKES

We illustrate how the error ratio can be used to identify the types of failures and strengths for the popular Vision Transformer (ViT) model. Despite impressive $84\%$ average top-1 accuracy, we find ViT's mistakes are associated with texture, occlusion, and subcategory (appearing 2.02-2.11x more times among misclassified samples than overall) as shown in Figure 5. On the other hand we find ViT are robust to pose, brightness, pattern, and partial views. We also see that these strenghts or weaknesses can vary by metalabel. For example, ViT is susceptible to occlusion for vessel and snake, but not the commodity metalables where mistakes are associated with style, darkness, and texture. For the dog metaclass, ViT is quite robust to different poses. Instead, ViT's mistakes for dogs are associated with the presence of multiple objects and differences among dog breeds (subcategory). The full list of failure types across meta-labels is in Appendix A.12.

### 4.3 WHICH LEVERS CAN AFFECT MODEL ROBUSTNESS?

In practice model developers have many choices from architecture, learning paradigm, and training regularization to data augmentations. What impact do each of these choices have on model robustness? We systematically examine how each choice affects robustness.

#### 4.3.1 ROLE OF SUPERVISION

We first group models into supervised (1k and with more data), self-supervised, and trained with robustness interventions in Figure 1. We measure the error ratio for each factor across in `ImageNet-X`. We find all model types have comparable error ratios, meaning models make similar types of mistakes. There are a few minor differences. For instance, self supervised models seem to be slightly more robust to the factors: color, larger, darker, style, object blocking, subcategory and texture. Supervised models trained on more data are more robust to the person blocking factor. We isolate whether some of the effects may be due to difference in data augmentation next.

#### 4.3.2 DATA AUGMENTATION

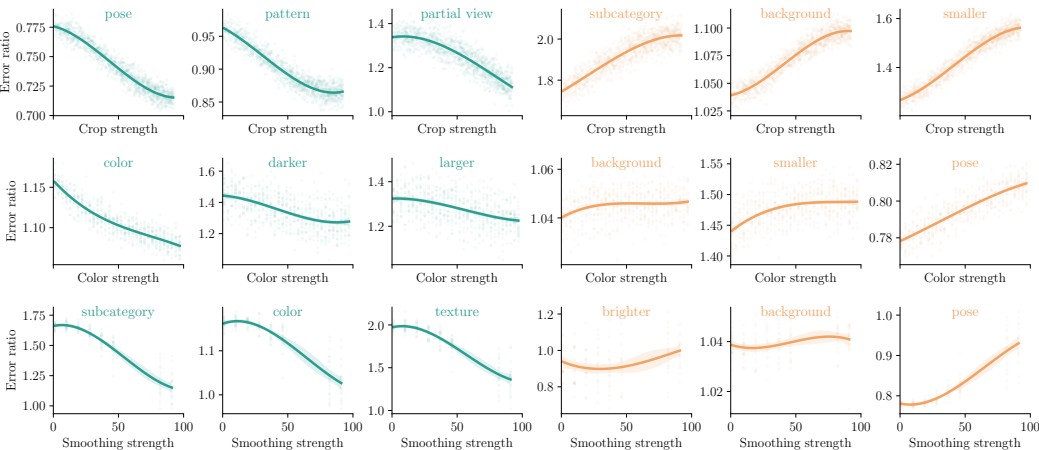

Figure 6: Evaluation of the impact of DA towards robustness of the trained model. We experiment with random crop (top row), color jittering (middle row) and Gaussian blur (bottom row); in all cases, we vary the strength of the respective DA (x-axis) and train multiple independent models for each setting. We observe that decreasing the random crop lower-bound (i.e. increasing the DA strength going from left to right) leads to more robust performance when pose varies, or partial views of the objects are present. Interestingly this also reduce the model robustness to objects that appear smaller and to background variations i.e. **the random crop augmented models become robust towards background and texture.** Color jittering improves robustness to darker objects and to objects with varying color. Surprisingly, color jittering also increases robustness to larger objects. Gaussian blur also presents a dual effect i.e. benefits towards subcategory and color changes and increases sensitivity to change in pose. In short, it seems that regardless of the DA, the benefits are always accompanied by detrimental impacts on unexpected factors hinting at a possible limitation of DA to produce improved robustness throughout all factors.

Data-augmentation (DA) is an ubiquitous technique providing significant average test performance gains (Shorten and Khoshgoftaar, 2019). In short, DA leverages a priori knowledge to artificially augment a training set size. The choice of DA policies e.g. image rotations, color jittering, translations is rarely questioned as it is given by our a priori understanding of which transformations will produce nontrivial new training samples whilst preserving their original semantic content. Although intuitive at first glance, DA has recently seen much attention as it can unfairly impact different classes (Balestriero et al., 2022), affect invariance (Deng et al., 2022), and potential to improve various forms of robustness (Hendrycks et al., 2021; Mintun et al., 2021). Equipped with `ImageNet-X`, we now propose a more quantitative and principled understanding of the impact of DA.

To that extend, we propose the following controlled experiment. We employ a ResNet50 —which is one of the most popular architecture employed by practitioners— and perform multiple independent training runs with varying DA policy. Each run across all policies share the exact same optimizer (SGD), weight-decay (1e-5), mini-batch size (512), number of epochs (80), and data ordering through training. For each DA setting, multiple runs are performed to allow for statistical testing. Only the strength of the DAs and the random seed vary within those runs. In all scenarios, left-right flip is used both during training and testing.

**Data augmentations can improve robustness, but with spill-over effects to unrelated factors.** We report in Fig. 6 the statistically significant effects on error ratio due to three data augmentations: random crop, color jittering and Gaussian blur. For each setting, we measure prevalence shifts i.e. how much more or less likely a factor is to appear among a models' misclassifications.

For random crop, we vary the random crop area lower-bound i.e. how small of a region can the augmented sample be resized from the original image (varying from $0.08$ which is the default value, to $1$ which is synonym of no DA). We find the prevalence shift decreases for pose and partial view as expected. We also find that pattern, which is unrelated to random cropping, improves as well. However, we also observe a decrease in robustness for subcategory, an unrelated factor.

For color jittering, we vary both the probability to apply some standard color transformations and their strength. In particular, for any given value $t \in (0, 1)$ we employ the composition of ColorJitter with probability $0.8t$ with brightness strength $0.4t$, contrast strength $0.4t$, saturation strength $0.1t$, and hue strength $0.1t$, followed by random grayscale with probability $0.5t$, and finally followed by random solarization with probability $0.3t$. Those parameters are found so that when $t$ nears $1$, the DA is similar to most aggressive DA pipelines used in recent methods. We naturally observe that the predictions become more robust to change in color and brightness. Interestingly, the model becomes more sensitive to pose variations.

For Gaussian blur, we vary the standard deviation of the Gaussian filter between $0.1$ and $0.1 + 3.5t$ for a filter size of $13 \times 13$. Note that when $t$ approaches $1$ the strength of this DA goes beyond the standard setting used, for example, in SSL. In supervised training, GaussianBlur DA is rarely employed on ImageNet. We observe that the model becomes more robust to texture which might come from blurring removing most of the texture information and forcing the model to rely on other features. Surprisingly, subcategory is also much less of an impact on performances while change in pose becomes more problematic for the Gaussian blur trained model.

## 5 CONCLUSION

We introduced `ImageNet-X`, an annotation of the validation set and 12,000 training samples of the ImageNet dataset across 16 factors including color, shape, pattern, texture, size, lightning, and occlusion. We showed how `ImageNet-X` labels can reveal how images in popular ImageNet dataset differ. We found that images commonly vary in pose and background, that classes can have distinct factors (such as dogs more often varying in pose compared to other classes), and that ImageNet's training and validation sets share similar distributions of factors. Next, we showed how models mistakes are surprisingly consistent across architectures, learning paradigms, training data size, and common robustness interventions. We identified data augmentation as a promising lever to improve models' robustness to related factors, however, it can also affect unrelated factors. These findings suggest a need for a deeper understanding of data augmentations on model robustness. We hope that `ImageNet-X` serves as an useful resource to build a deeper understanding of the failure modes of computer vision models, and as a tool to measure their robustness across different environments.

**Reproducibility Statement** The results and figures in the paper can be reproduced using the open-source code and the `ImageNet-X` annotations, which we also release. The annotations were collected by training annotators to contrast three prototypical images from the same class. This setup that can be replicated using the questionnaire we provide here as well as the freely available ImageNet dataset. For a detailed description of the `ImageNet-X` dataset, please see A.1.

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

# A APPENDIX

We include additional experiments and dataset details in this appendix. In the first several sections, we provide additional information regarding the ImageNetX dataset: annotation details in A.1, factor definitions in A.2, number of factors selected in A.3, and factor co-occurence in A.4. Finally, we show sample annotations in Appendix A.5.

Next, we show additional experiments. First in A.6 we show details of ImageNet-X robustness analysis for additional architectures and learning paradigms. We show how such comparisons can inform modeling choices in A.10. In Appendix A.7 we detail our experiments in applying tailored augmentations per class and show in A.8 how label bias measured in ImageNet-ReaL correspond to ImageNet-X factors. Next we show how ImageNet-X factors can probe downstream performance on OOD datasets in Appendix A.9. Then, we replicate our analysis using multi-factor labels in A.11. Finally, in A.12 we illustrate how ImageNet-X factors can reveal granular model robustness strengths and weaknesses for specific class categories.

## A.1 DATA SHEET

We follow the recommendations from by constructing a datasheet (Gebru et al., 2021) for `ImageNet-X` below.

**For what purpose was the dataset created?** We created `ImageNet-X` to enable research on understanding the biases and mistakes of computer vision classifiers on the popular ImageNet benchmark.

**What do the instances that comprise the dataset represent?** Each instance represent a set of binary labels and free-form text descriptions of variation for each ImageNet validation image.

**How many instances are there in total?** There are 50,000 validation and randomly sampled 12,000 training image labels.

**What data does each instance consist of** Each data point is a json string indicated whether each of the 16 factors was selected and two free-form text fields.

**Are there any errors, sources of noise, or redundancies in the dataset?** Each sample was labeled by a single human annotator. There are no other redundancies or sources of errors.

**Is the dataset self-contained?** The full annotations each corresponding ImageNet filename will be provided in a self-contained repo. The original ImageNet images however, must be downloaded separately.

**Does the dataset identify any subpopulations (e.g., by age, gender)?** No.

**Is the software that was used to preprocess/clean/label the data available?** Yes, the data was preprocessed using Pandas and Numpy, both freely available Python packages.

## A.2 FACTOR OF VARIATION DEFINITIONS

| Factor of variation | Description |
| --- | --- |
| Pose | The object has a different pose or is placed in different location within the image. |
| Partial view | The object is visible only partially due to the camera field of view that did not contain the full object – e.g. cropped out. |
| Object blocking | The object is occluded by another object present in the image. |
| Person blocking | The object is occluded by a person or human body part – this might include objects manipulated by human hands. |
| Multiple objects | There is, at least, one another prominent object present in the image. |
| Smaller | Object occupies only a small portion of the entire scene. |
| Larger | Object dominates the image. |
| Brighter | The lighting in the image is brighter when compared to the prototypical images. |
| Darker | The lightning in the image is darker when compared to the prototypical images. |
| Background | The background of the image is different from backgrounds of the prototypical images. |
| Color | The object has different color. |
| Shape | The object has different shape. |
| Texture | The object has different texture – e.g., a sheep that's sheared. |
| Pattern | The object has different pattern – e.g., striped object. |
| Style | The overall image style is different– e.g., a sketch. |
| Subcategory | The object is a distinct type or breed from the same class – e.g., a mini-van within the car class. |

## A.3 NUMBER OF IMAGES WITH X ACTIVE FACTORS

Figure 7 shows the count of samples depending on how many factors are selected by the annotator.

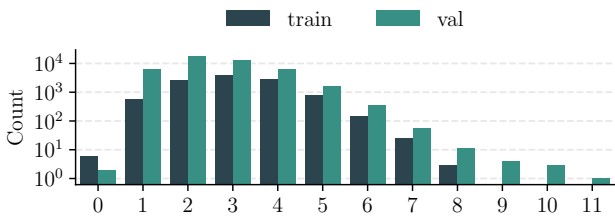

Figure 7: **Many images have multiple factors selected. Very few have no factors selected since some factors are sensitive** Distribution of the per image number of factors annotated as active, in validation vs train.

## A.4 FACTOR CO-OCCURRENCES AND CORRELATIONS.

Table 2 reports the exact count of each factor. Furthermore, Figure 8 (left) shows the spearman correlation between each factor and meta-class and (right) between each pair of factors for the validation set. Figure 9 shows the same for the training set for comparison. The major correlations are the same between the two heatmaps (for instance pattern and color or pattern and dog). We can see that some factors are slightly correlated, for example color and pattern, larger and partial view, subcategory and shape are positively correlated.

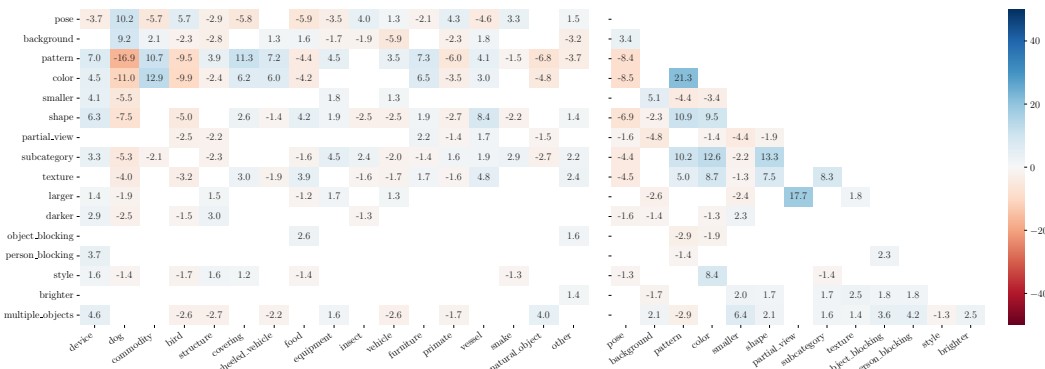

Figure 8: Factor/factor and factor/meta-label spearman correlation heatmaps for validations set. The heatmap shows only significant correlations (p value $\leq 0.05$).

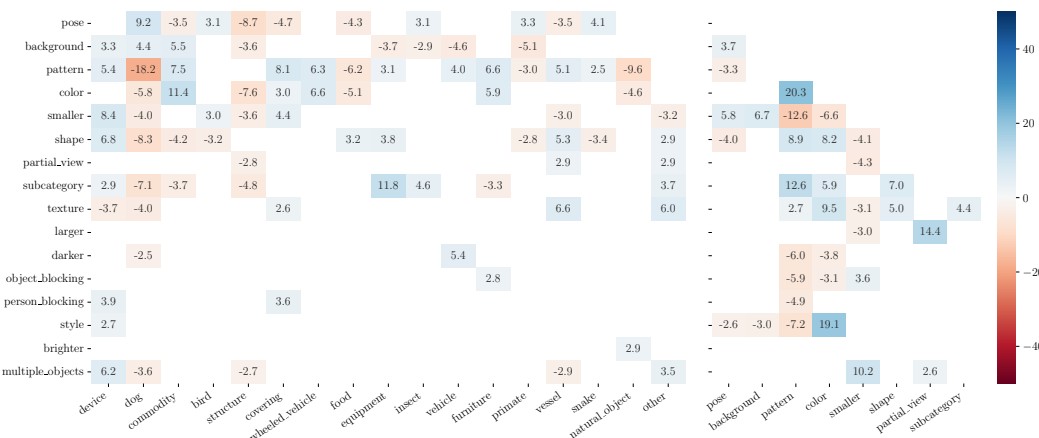

Figure 9: Factor/factor and factor/meta-label spearman correlation heatmaps for training set. The heatmap shows only significant correlations (p value $\leq 0.05$).

| Split | Factor selection | factor | count | factor | count | factor | count | factor | count |
|---|---|---|---|---|---|---|---|---|---|
| Validation | All factors | pose | 39862 | background | 35467 | pattern | 15401 | color | 12995 |
| | | subcategory | 3606 | smaller | 2866 | shape | 1889 | multiple objects | 1837 |
| | | partial view | 1504 | texture | 878 | style | 574 | darker | 515 |
| | | larger | 390 | object blocking | 219 | person blocking | 105 | brighter | 95 |
| | Top factor | pose | 15031 | background | 14743 | color | 6138 | pattern | 5938 |
| | | smaller | 1449 | shape | 675 | partial view | 640 | subcategory | 586 |
| | | texture | 283 | larger | 151 | darker | 123 | object blocking | 80 |
| | | person blocking | 60 | brighter | 45 | style | 44 | multiple objects | 41 |
| Training | All factors | pose | 10349 | background | 9618 | pattern | 6871 | color | 3750 |
| | | smaller | 1192 | subcategory | 717 | shape | 514 | multiple objects | 462 |
| | | partial view | 362 | style | 357 | texture | 293 | darker | 99 |
| | | larger | 82 | object blocking | 69 | person blocking | 34 | brighter | 7 |
| | Top factor | pose | 3250 | background | 2936 | pattern | 2525 | color | 1438 |
| | | smaller | 494 | subcategory | 128 | shape | 127 | partial view | 118 |
| | | texture | 93 | larger | 35 | darker | 26 | style | 17 |
| | | object blocking | 17 | person blocking | 16 | multiple objects | 5 | brighter | 4 |

Table 2: Counts of each factor of variation in `ImageNet-X`.

## A.5 EXAMPLE ANNOTATIONS

In Figure 11 we present some sample annotations. We show the prototypical images for each class along with the sample annotators labeled.

(a) Factor/factor co-occurrence in the training subset.

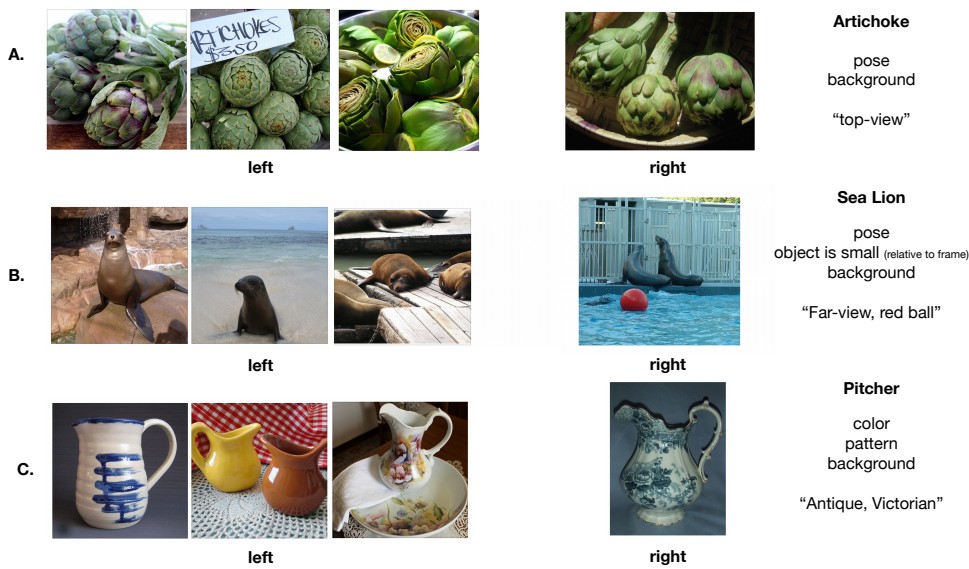

Figure 11: **ImageNet-X Annotations Examples** illustrates three annotation examples from the ImageNet validation set. In panel, A. we see an example of background and pose for Artichoke. In panel B., we see the effect of an object being small and in panel C. we see color and pattern as distinctive varying factors. We provide the full set of annotations with their image paths in our repo.

## A.6 RESULTS ON POPULAR MODELS AND METHODS

### A.6.1 ASSESSING THE ROBUSTNESS OF VISION TRANSFORMERS

To move beyond average accuracy, we can probe model robustness with varying levels of granularity by measuring bias towards or against ImageNet-X factors. Here we walk through the biases of a ViT (vision transformer model) trained on ImageNet-21k. While the average accuracy is $> 80\%$, we probe further to uncover ViT biases towards texture, occlusion, and subcategory.

**Assessing ViT's bias modes**  We first examine the prevalence shift of each factor among a models' misclassified samples relative to their overall prevalence. Despite impressive average top-1 accuracy, we find ViT is susceptible to several factors with texture, occlusion, and subcategory appearing +1.02-1.11x more times among misclassified samples than overall as shown in Figure 12. On the other hand we find ViT are robust to naturally occurring pose, brightness, pattern, and partial views.

**Granular biases by meta-label**  We further probe robustness by examining how model bias differs among meta-labels of classes groups. We find, as shown in Figure 13, the three meta-labels with the lowest accuracies are vessel, snake, and commodity. We find ViT is most bias towards occlusion (+2.0x-2.6x) for vessel and snake and style for commodity. While ViT is robust to darkness and brightness (-1.0x) for vessel, it's susceptible to darkness (+1.2-1.6x) for snakes and the commodity meta-labels. Among these three meta-labels ViT is susceptible to shape (+1.4x) for the snake meta-label. Model bias across all 16 meta-labels is in Appendix Figure 12.

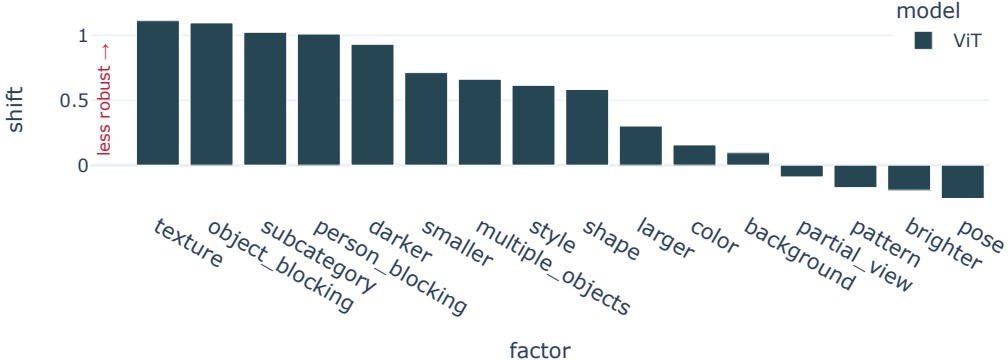

Figure 12: ViT model bias

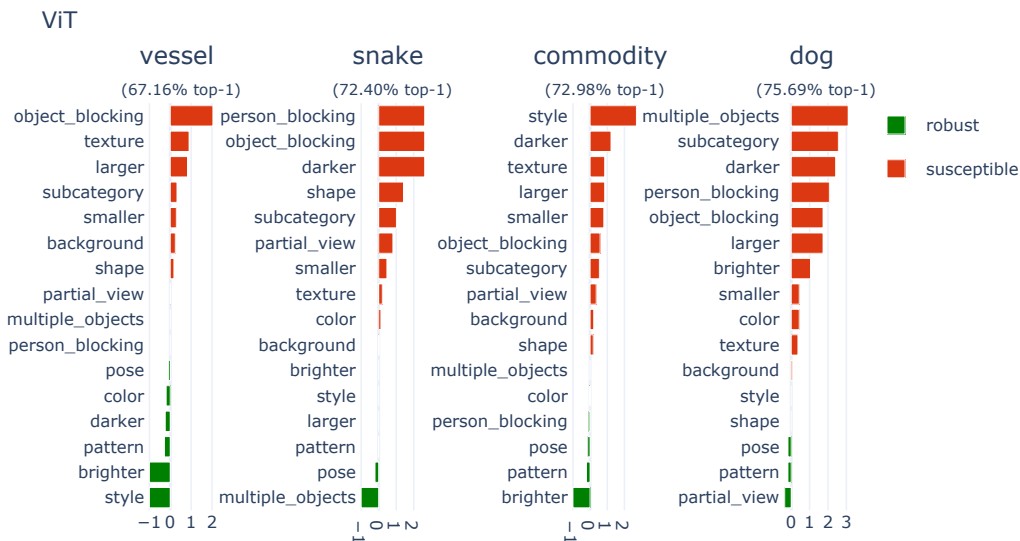

Figure 13: **Robustness by meta-label** shows robustness of the three worst meta-labels for ViT (vision transformer) as well as the dog class which encompaces a large number of ImageNet classes.

### A.6.2 Architecture and learning paradigm

**Convolution and transformer-based models exhibit similar overall biases** : As shown in Figure 14, ResNet-50 and ViT exhibit similar prevalence in factors among their misclassified samples suggesting both models exhibit robustness to similar factors. Both ViT and ResNet-50 show a bias among misclassifications towards texture, occlusion by a person or object, subcategory, darkness and the prescence of multiple objects. For example, texture is more likely to appear among misclassified compared to the overall validation set by +1.48x for ResNet-50 and +1.1x for ViT. This confirms finding in Geirhos et al. (2018) illustrating CNNs are bias towards texture. We extend this finding by illustrating transformer based models are also biased towards texture. The overall similarity between CNN and transformer based models matches the findings in Bouchacourt et al. (2021) for invariance to augmentations by illustrating the similarity holds even for natural factors of variation. While the

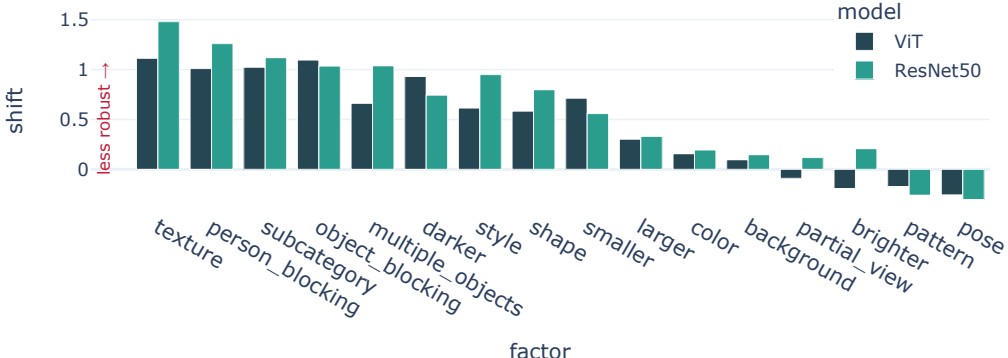

Figure 14: **Comparison of factor error ratios for ViT (transformer-based) and ResNet-50 (CNN-based) models.** The y-axis measures the shift in how likely a factor is to appear in a model's misclassified samples relative to their overall prevalence. A shift greater than zero measures the likelihood a factor is model bias towards a factor suggesting a model is susceptible to the factor. A shift below zero implies a factor is less likely to appear among misclassified suggesting the model is robust to the factor.

overall trends are similar we observe the largest difference between ViT and ResNet-50 in their biases towards texture, the presence of multiple objects, and style with ViT exhibiting a smaller bias.

**Role of learning supervision is important for SimCLR versus ResNet-50, but not transformer based ViT versus DINO** : In Figure 15b, we compare the factor bias in self-supervised versus supervised learning for both ResNet-based and transformer-based models. We observe remarkably similar biases when comparing ViT versus DINO in 15a. For ResNet based models, we observe self-supervised learning in SimCLR leads to less bias towards the presence of multiple objects, style, and darkness. Specifically, for a supervised ResNet-50 the prevalence with multiple objects is +1.03x (versus +0.22x for SimCLR), among misclassified samples. This suggests while learning paradigm leads to simlar biases for transformers, standard supervised training does lead to differences in bias towards texture, multiple objects, and darkness. We isolate the effect of data augmentation in next Section.

*A more granular comparison of learning paradigm for transformers by meta-label*: We compare the worst meta-labels for ViT versus DINO, finding both more share the worst three meta-labels with remarkably similar biases: occlusion and texture for vessel, style and darkness for commodity, and occlusion for snake. There is a difference in the magnitude of some factors. For example, DINO is more susceptible to partial views for snakes compared to ViT (+1.76x versus +0.81x). Aside from minor differences, the granular meta-labels analysis confirms DINO and ViT exhibit similar robustness for their worst meta-labels.

## A.7 IMPROVING TRAINING

### A.7.1 TAILOR AUGMENTATIONS BY CLASS

To test whether tailoring augmentations by class improves robustness, we identify meta-labels benefiting from the robustness induced by two types of augmentations: color jitter and random resized crop. For color jittering, we select meta-labels with a prevalence shift ¿ 1.0 for either darker, color, or larger factors (whose robustenss improves with color jittering). The resulting 402 classes belonging to those meta-labels are then augmented with additional color jittering augmentations on top of the standard training recipe with random resized cropping. We show results against baselines of only cropping, only color jittering, and both augmentations applied to all classes in Figure 17.

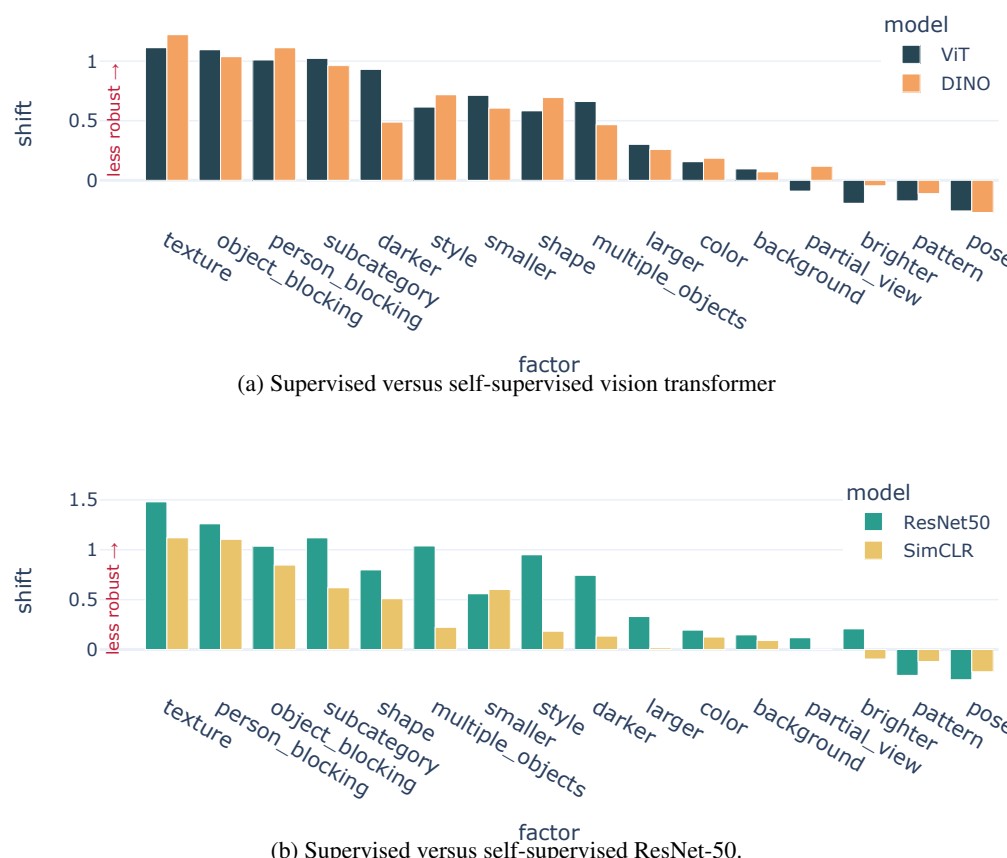

(a) Supervised versus self-supervised vision transformer

(b) Supervised versus self-supervised ResNet-50.

Figure 15: **Comparison of error ratios across self-supervised and supervised learning.** The y-axis measures the shift in how likely a factor is to appear in a model's misclassified samples relative to their overall prevalence. A shift greater than zero measures the likelihood a factor is model bias towards a factor suggesting a model is susceptible to the factor. A shift below zero implies a factor is less likely to appear among misclassified suggesting the model is robust to the factor.

## A.8   EVALUATING LABEL BIAS

In Figure 18 we measure the ReAL accuracy relative to the original validation accuracy.

## A.9   CORRELATION WITH IMAGENET DISTRIBUTION SHIFTS

Imagenet-X provides fine grained labels to evaluate model robustness. To link the performance of this work with related work in Imagenet robustness, we provide a pearson correlation score between factor error ratios and distribution shift error ratios in figure 19.

| ImageNet-R | ObjectNet-1.0-beta | ImageNet-A | Greyscale | ImageNet-Vid-Robust_pm0 |
|---|---|---|---|---|
| texture (0.93) | texture (0.98) | texture (0.96) | color (0.79) | texture (0.77) |
| shape (0.90) | subcategory (0.97) | shape (0.96) | smaller (0.74) | smaller (0.77) |
| subcategory (0.90) | shape (0.96) | smaller (0.94) | person_blocking (0.72) | style (0.77) |

Table 3: We show the Pearson correlation (shown in parenthesis) between factor error ratios and performance on distribution shifts of ImageNet for the 3 most correlated factors.

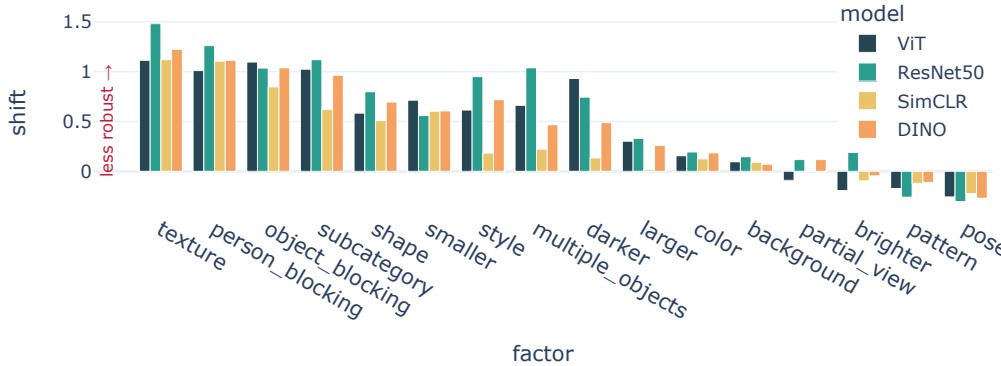

Figure 16: **Models tend to have similar biases for the measured factors. The largest difference being between ResNet and SimCLR likely to color jitter.**

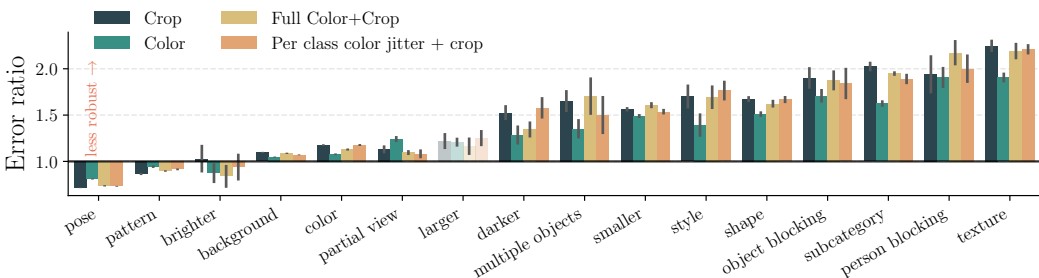

Figure 17: **Prevalence shift comparison of tailored versus universal color jittering augmentation**

In Table 3, we show the three most correlated factors with each OOD dataset. We compute the Pearson correlation between each factor's error ratio (shown in parenthesis) and the performance on each OOD dataset. We find an intuitive correspondence between the factors in ImageNet-X and downstream OOD performance. For example, ImageNet-R, which contains renditions such as art, cartoons, embroidery, graphics, origami, paintings, patterns of ImageNet classes, is most correlated with shape and texture as well as subcategory. On the other hand, Greyscale which discards color information is most influence by each model's reliance on color, smaller objects, and occlusion. ImageNet-A, which contains natural images explicitly mined to be challenging for standard vision models, often contains images where the object is small or blends in with the background. We see the top factors for ImageNet-A are texture, shape, and whether an object can be identified even if it's small. We hope this analysis reveals the potential for ImageNet-X factors to better probe downstream performance. This unlocks for example the potential for future work to assess/optimize downstream OOD performance only by tuning model performance on the ImageNet-X factors.

## A.10   INFLUENCE OF ARCHITECTURE CHOICES

Although the general conclusion of our per factor performance analysis indicates consistent performances across architectures, training data and losses, there are still some differences that arise. We study some model choices to shed some light on where those differences might arise in Figures 20, 21 and 22.

## A.11   MULTI FACTOR RESULTS

In the paper we chose to select the top factor per image. In this section we present the main figures of the paper with multiple factors per image selected. The main conclusions remain valid, and the

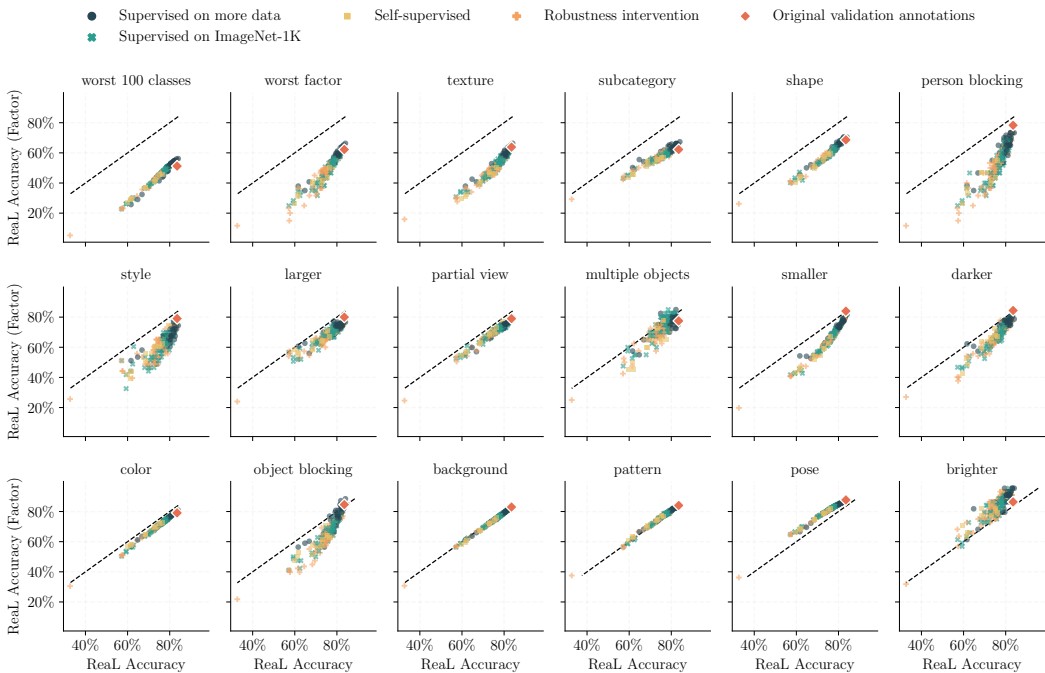

Figure 18: We measure the ReAL accuracy Beyer et al. (2020b) of the original validation labels and show that these labels have consistent errors on the same factors as models shown previously.

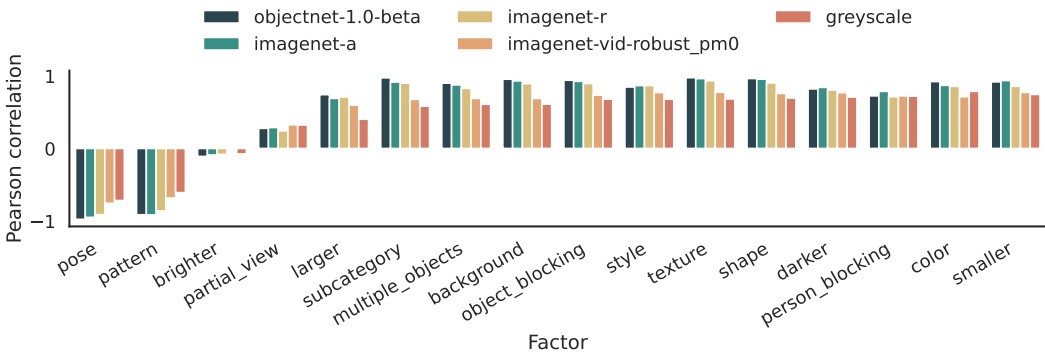

Figure 19: Pearson correlation between factor error ratios and distribution shifts of ImageNet. The correlation strengths change depending on the distribution shift. For instance for greyscale images, the color factor is maximally correlated with relative performance.

results don't drastically change. We include the multifactor setting for completeness. The provided code allows generating figures and loading annotations both in top and multi factor settings. See figures 23 and 24

## A.12 GRANULAR MODEL BIASES BY META-LABELS

In Table 4 we show the robustness susceptibility of factors for DINO, ResNet50, SimCLR and ViT for each meta-label. We see while many factors are shared, some meta-labels are especially susceptible to distinct factors. For example, ViT is susceptible to style for vehicles and insects.

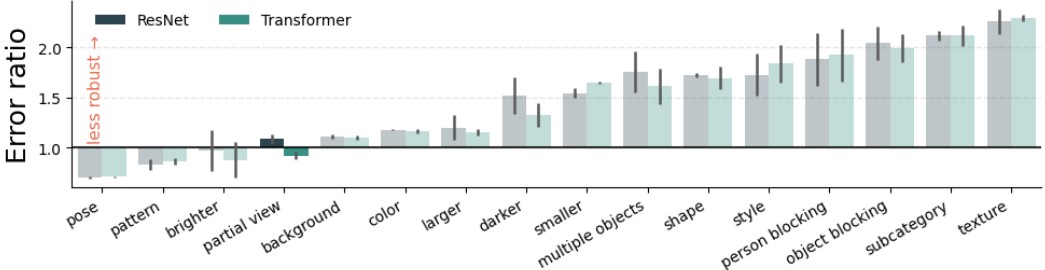

Figure 20: We measure the error ratios for multiple resnets (ResNet50, ResNet101, ResNet152) and for multiple Transformers (ViT small/16, base/16, large/16). The main differences are in the factors partial view, darker and multiple objects, with transformers having an advantage. The better handling of multiple objects in Transformers might be due to the lack of average pooling.

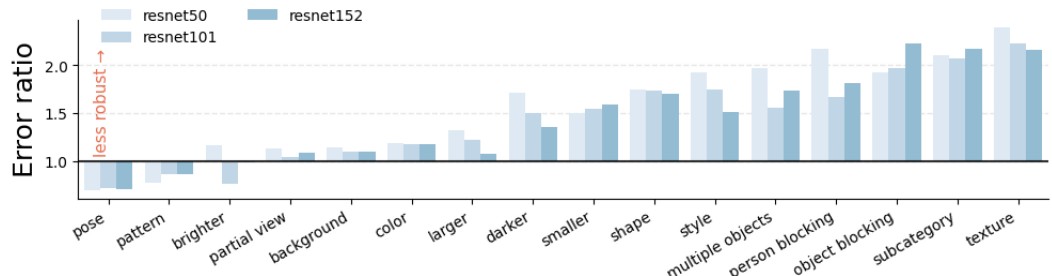

Figure 21: We measure the error ratios for multiple resnets (ResNet50, ResNet101, ResNet152) to study the effect of depth. For most factors, the deeper the resnet the better, except for the "object blocking" factor that gets considerably worse as the model gets deeper.

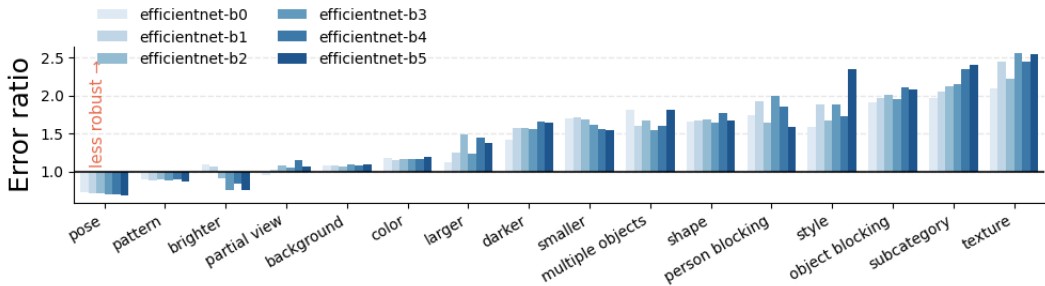

Figure 22: We measure the error ratios for 6 scales of EfficienNet models from b0 to b5. As the model grows in size, the error ratios grow, especially for the b5 scale.

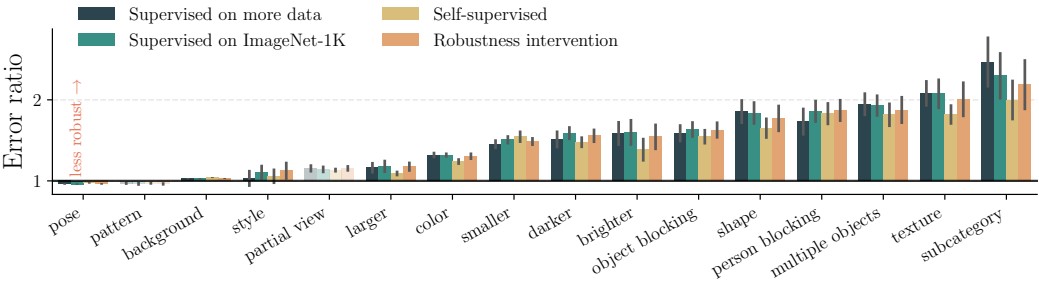

Figure 23: Model comparison with multiple factors per image.

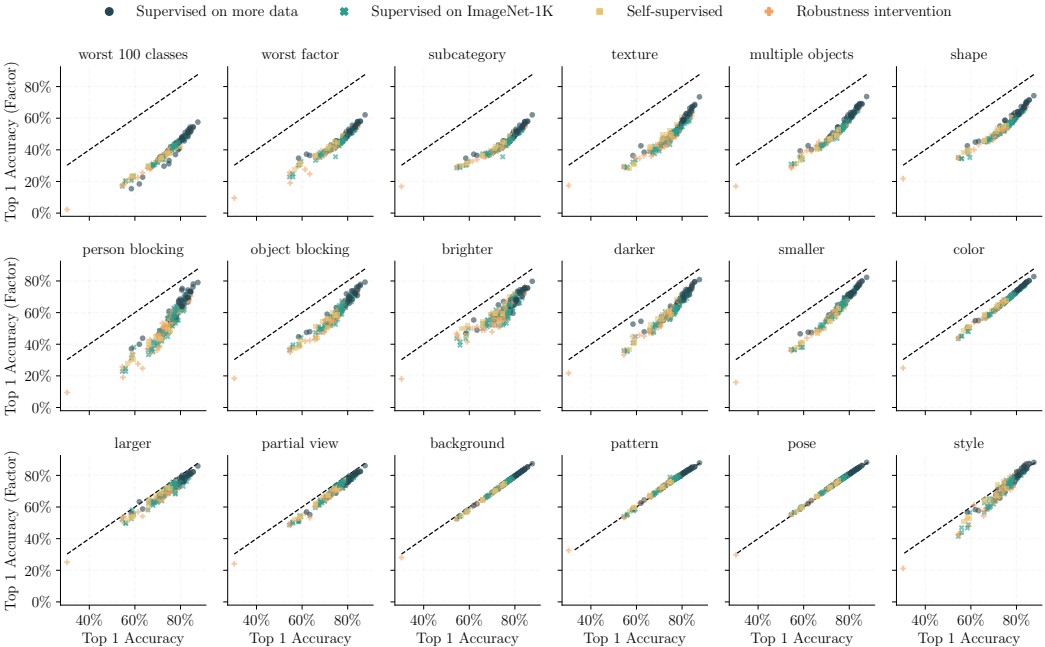

Figure 24: High level scatter plot with multiple factors per image.

| Model | Metaclass | Robust | Susceptible |
|---|---|---|---|
| DINO | bird | person blocking (0.0), style (0.0), texture (0.0), larger (0.0) | darker (2.6), smaller (2.7), object blocking (12.9), shape (12.9) |
| | commodity | brighter (0.0), texture (0.7), pattern (0.8), pose (0.8) | partial view (1.7), darker (2.0), object blocking (2.3), style (3.3) |
| | covering | pose (0.8), larger (0.8), pattern (0.9), partial view (0.9) | texture (1.6), smaller (1.6), style (2.5), object blocking (2.7) |
| | device | partial view (0.7), pattern (0.8), style (0.8), brighter (0.8) | smaller (1.5), person blocking (1.6), texture (2.3), object blocking (2.7) |
| | dog | pattern (0.7), pose (0.8), partial view (0.9), smaller (0.9) | object blocking (3.0), subcategory (3.1), person blocking (4.5), multiple objects (4.5) |
| | equipment | darker (0.0), style (0.0), larger (0.3), pattern (0.7) | smaller (1.8), object blocking (1.9), subcategory (2.2), person blocking (3.8) |
| | food | style (0.0), larger (0.0), brighter (0.0), pose (0.5) | darker (1.5), partial view (1.6), subcategory (2.6), texture (2.6) |
| | furniture | object blocking (0.0), larger (0.0), style (0.0), multiple objects (0.0) | shape (1.4), subcategory (1.6), smaller (1.9), darker (3.3) |
| | insect | texture (0.0), larger (0.0), multiple objects (0.0), brighter (0.0) | shape (3.3), style (3.3), subcategory (4.2), darker (6.6) |
| | natural object | brighter (0.0), pose (0.5), partial view (0.9) | larger (2.7), darker (2.7), subcategory (3.2), object blocking (5.3) |
| | other | pose (0.7) | subcategory (2.0), multiple objects (2.0), style (2.6), person blocking (3.0) |
| | primate | partial view (0.0), larger (0.0), style (0.0), pose (0.7) | object blocking (2.5), subcategory (5.1), shape (5.1), texture (5.1) |
| | snake | darker (0.0), multiple objects (0.0), pose (0.8), pattern (1.0) | subcategory (2.2), partial view (2.9), person blocking (3.3), object blocking (3.3) |
| | structure | object blocking (0.0), brighter (0.0), darker (0.6), color (0.7) | person blocking (1.8), subcategory (2.4), style (2.4), texture (2.6) |
| | vehicle | person blocking (0.0), texture (0.0), multiple objects (0.0), pattern (0.5) | smaller (2.0), style (3.0), darker (3.0), shape (3.0) |
| | vessel | brighter (0.0), pattern (0.8), pose (0.9), color (1.0) | larger (1.5), style (1.7), texture (1.8), object blocking (2.6) |
| | wheeled vehicle | object blocking (0.0), person blocking (0.0), texture (0.0), style (0.0) | shape (1.6), smaller (1.6), darker (2.4), larger (4.9) |
| ResNet50 | bird | larger (0.0), style (0.0), multiple objects (0.0), brighter (0.0) | darker (4.8), object blocking (11.9), person blocking (11.9), shape (11.9) |
| | commodity | pattern (0.6), person blocking (0.8), pose (0.8) | texture (2.2), larger (2.6), brighter (3.1), style (3.1) |
| | covering | pattern (0.7), larger (0.8), pose (0.8), person blocking (0.9) | smaller (1.5), texture (1.7), object blocking (2.0), style (3.3) |
| | device | pattern (0.6), partial view (0.7), pose (0.8), color (1.0) | multiple objects (1.7), texture (2.3), object blocking (2.3), brighter (2.5) |
| | dog | pattern (0.4), texture (0.8), pose (0.8), partial view (0.8) | object blocking (3.0), subcategory (4.2), person blocking (4.6), multiple objects (4.6) |
| | equipment | darker (0.0), style (0.0), larger (0.3), pattern (0.5) | smaller (1.8), object blocking (1.9), subcategory (2.1), person blocking (3.9) |
| | food | larger (0.0), style (0.0), brighter (0.0), pose (0.4) | shape (1.5), subcategory (2.9), darker (2.9), texture (2.9) |
| | furniture | object blocking (0.0), larger (0.0), multiple objects (0.0), style (0.8) | smaller (1.6), texture (1.7), person blocking (2.0), darker (3.1) |
| | insect | brighter (0.0), pose (0.6), pattern (0.8), larger (1.0) | shape (3.6), subcategory (4.2), multiple objects (5.8), darker (5.8) |
| | natural object | brighter (0.0), pattern (0.2), pose (0.5) | darker (2.9), larger (2.9), subcategory (4.6), object blocking (5.7) |
| | other | pose (0.6), pattern (0.9) | subcategory (2.2), texture (2.3), multiple objects (2.3), person blocking (3.5) |
| | primate | partial view (0.0), larger (0.0), pose (0.8) | style (4.4), subcategory (4.4), shape (4.4), texture (4.4) |
| | snake | darker (0.0), multiple objects (0.0), pose (0.8), pattern (0.9) | texture (2.1), shape (2.1), object blocking (3.2), person blocking (3.2) |
| | structure | brighter (0.0), pattern (0.7), pose (0.8), color (2.3) | texture (2.0), style (2.3), subcategory (2.8), multiple objects (2.9) |
| | vehicle | person blocking (0.0), texture (0.0), multiple objects (0.0), pattern (0.5) | subcategory (2.4), shape (3.0), style (3.0), darker (4.4) |
| | vessel | pattern (0.7), pose (0.9), subcategory (0.9), shape (1.0) | style (1.6), larger (1.9), object blocking (2.4), brighter (2.4) |
| | wheeled vehicle | texture (0.0), brighter (0.0), style (0.0), pattern (0.4) | shape (2.5), object blocking (2.5), larger (3.4), person blocking (5.0) |
| SimCLR | bird | person blocking (0.0), texture (0.0), larger (0.0), multiple objects (0.0) | partial view (1.8), smaller (1.8), shape (5.7), object blocking (5.7) |
| | commodity | object blocking (0.7), pattern (0.8), shape (0.8), pose (0.9) | smaller (1.5), texture (1.7), style (2.5), brighter (2.5) |
| | covering | larger (0.7), partial view (0.7), pattern (0.8), pose (0.9) | smaller (1.8), texture (2.0), style (2.1), object blocking (2.2) |
| | device | larger (0.7), partial view (0.8), pattern (0.8), pose (0.9) | person blocking (1.7), brighter (1.9), texture (2.1), object blocking (2.5) |
| | dog | multiple objects (0.0), partial view (0.5), pose (0.8) | object blocking (1.9), darker (2.4), subcategory (2.7), person blocking (2.9) |
| | equipment | darker (0.0), style (0.0), pattern (0.6), multiple objects (0.8) | object blocking (1.5), subcategory (1.8), texture (2.7), person blocking (3.0) |
| | food | darker (0.0), larger (0.0), style (0.0), pose (0.6) | smaller (1.6), object blocking (2.0), subcategory (2.1), texture (2.2) |
| | furniture | object blocking (0.0), larger (0.0), multiple objects (0.0), style (0.7) | smaller (1.9), texture (2.3), subcategory (2.7), darker (2.7) |
| | insect | brighter (0.0), partial view (0.6), larger (0.7), pose (0.7) | subcategory (2.8), shape (2.9), multiple objects (3.9), darker (3.9) |
| | natural object | brighter (0.0), pose (0.6), background (1.0) | larger (2.8), subcategory (3.0), darker (3.7), object blocking (3.7) |
| | other | pose (0.7), brighter (0.8) | subcategory (1.7), shape (1.7), texture (1.9), person blocking (2.7) |
| | primate | larger (0.0), style (0.0), partial view (0.5), darker (0.8) | smaller (2.2), subcategory (3.3), texture (3.3), shape (3.3) |
| | snake | darker (0.0), multiple objects (0.0), texture (0.8), pose (0.8) | subcategory (1.6), partial view (2.2), person blocking (2.5), object blocking (2.5) |
| | structure | brighter (0.0), darker (0.5), color (0.8), pattern (0.8) | subcategory (1.5), person blocking (1.5), subcategory (2.0), texture (2.2) |
| | vehicle | person blocking (0.0), texture (0.0), multiple objects (0.0), pattern (0.7) | object blocking (1.9), darker (2.4), style (2.4), shape (2.4) |
| | vessel | brighter (0.0), darker (0.5), style (0.6), pattern (0.9) | smaller (1.2), partial view (1.4), texture (1.5), object blocking (1.9) |
| | wheeled vehicle | object blocking (0.0), brighter (0.0), pattern (0.5), shape (0.6) | larger (2.5), person blocking (3.8), style (3.8), texture (3.8) |
| ViT | bird | person blocking (0.0), object blocking (0.0), larger (0.0), texture (0.0) | subcategory (2.2), darker (2.7), smaller (2.7), shape (13.4) |
| | commodity | brighter (0.0), pattern (0.8), pose (0.9), person blocking (0.9) | texture (1.8), larger (1.8), darker (2.1), style (3.5) |
| | covering | person blocking (0.5), pattern (0.7), partial view (0.8), pose (0.9) | subcategory (1.7), smaller (2.0), style (2.7), object blocking (2.9) |
| | device | brighter (0.0), pattern (0.7), partial view (0.7), pose (0.9) | darker (1.6), smaller (1.6), multiple objects (1.9), object blocking (2.6) |
| | dog | partial view (0.6), pose (0.8), pattern (0.8) | darker (3.3), subcategory (3.4), multiple objects (3.9), person blocking (3.9) |
| | equipment | darker (0.0), texture (0.0), larger (0.7), pattern (0.7) | smaller (1.9), subcategory (1.9), object blocking (2.0), person blocking (4.1) |
| | food | larger (0.0), brighter (0.0), style (0.0), pose (0.5) | darker (1.7), smaller (1.8), subcategory (2.3), texture (2.4) |
| | furniture | object blocking (0.0), larger (0.0), multiple objects (0.0), style (0.0) | texture (1.7), subcategory (2.0), smaller (2.1), darker (4.1) |
| | insect | texture (0.0), brighter (0.0), multiple objects (0.0), pose (0.7) | larger (2.4), subcategory (3.2), style (7.1), darker (7.1) |
| | natural object | partial view (0.0), pattern (0.6), pose (0.6), color (1.0) | brighter (2.2), subcategory (2.7), object blocking (3.4), darker (3.4) |
| | other | pose (0.7), brighter (0.8) | shape (1.8), texture (2.3), subcategory (2.3), person blocking (3.6) |
| | primate | larger (0.0), partial view (0.0), pose (0.7) | darker (4.6), texture (4.6), style (4.6), subcategory (4.6) |
| | snake | multiple objects (0.0), pose (0.8) | shape (2.3), person blocking (3.4), object blocking (3.4), darker (3.4) |
| | structure | person blocking (0.0), pattern (0.7), partial view (0.8), color (0.8) | texture (2.1), subcategory (2.3), style (2.5), multiple objects (2.5) |
| | vehicle | multiple objects (0.0), person blocking (0.0), texture (0.0), shape (0.0) | subcategory (1.4), darker (1.8), smaller (2.3), style (3.5) |
| | vessel | brighter (0.0), style (0.0), darker (0.7), pattern (0.7) | subcategory (1.3), larger (1.7), texture (1.8), object blocking (2.9) |
| | wheeled vehicle | shape (0.0), brighter (0.0), style (0.0), pattern (0.6) | object blocking (2.6), larger (3.5), person blocking (5.2), texture (5.2) |

Table 4: Best and worst 4 factors for DINO, ResNet50, SimCLR and ViT by Metaclass. The values between parenthesis are the error ratios for the associated factor.

