# OpenReview forum: "ImageNet-X: Understanding Model Mistakes with Factor of Variation Annotations"
_ICLR.cc/2023/Conference — ICLR 2023 notable top 25%_

### Official Review · Reviewer_8YJy · 2022-10-22

**Confidence:** 4
**Correctness:** 3
**Technical Novelty And Significance:** 3
**Empirical Novelty And Significance:** 3
**Recommendation:** 8

**Clarity, Quality, Novelty And Reproducibility:**

The overall presentation quality is good but I would suggest to move related work section after introduction to highlight the difference between this work and previous ones. Also, the paper states 2200 trained models are included in the experiment study but later only 209 models are used to create plots? Please clarify this.

The release of ImageNet-X and the corresponding experiments will be helpful for other model development and data usages.

**Strength And Weaknesses:**

Strengths:

- This work is well motivated and presented. It proposes a systematic way to label dominant variation type of an instance image against the corresponding canonical class pattern. This is in turn used to provide a concrete picture of model error modes in different classes and across different methods.

- A large number of training models have been evaluated in the new benchmark and some technical insights have been revealed quantitatively, though previous work is already aware of many empirical observations (e.g. Figure 6).

Weaknesses:

- It is not clear whether the additional variation factor label can benefit training and further improve the generalization of existing models. The authors have also labeled some training images but they did not show how these newly labeled training images could boost performance.

- In general, some data is hard to be labeled with only a single dominant factor or it can be perceived differently by different editors. This work did not discuss in detail about how to resolve label inconsistency and how to utilize multi-factor setting generally.

**Summary Of The Paper:**

This paper presents an enhanced ImageNet validation dataset (ImageNet-X) with a more detailed labeling about variation factors of an object instance against its class. Each validation image has been labeled by human editor with 16 different variation factors including pose, background, color and etc.. Hundreds of image classification models have been evaluated on this new enhanced dataset to provide a detailed view towards model failure modes in terms of these variation factors.

**Summary Of The Review:**

I think this paper presents a scientific point of view to analyze existing recognition methods and contributes a useful enhanced ImageNet variant for a potentially more diverse purpose. Some critical questions (in weakness section) related to using this new approach/dataset may be also discussed/addressed in detail to solidify the value of this work.

-- Post Rebuttal --

The authors' responses address most of my concerns. The new experiment results are good examples to show useful applications downstream. It would be nice that the authors can move this new additional results to the main paper. Given the current form of the submission, I am leaning towards acceptance.

---

> ### Author Response · Authors · 2022-11-15
> **Response**
>
> We appreciate you found our work well motivated, presented, and systematic. Based on your suggestion, we’ve also moved the related work earlier to improve the clarity. We provide respond to each question below:
>
> > “It is not clear whether the additional variation factor label can benefit training and further improve the generalization of existing models. The authors have also labeled some training images but they did not show how these newly labeled training images could boost performance.”
>
> We thank the reviewer for pointing this out. To address this shortcoming, we now include two additional experiments to illustrate how ImageNet-X factor labels can serve as a tool for model selection and informing choices such as architecture.In Appendix A.10, we add a detailed comparison of how architectural choices such as model depth and layer blocks (convolutional versus transformer-based) affect model robustness across factors. Next, we also show in Appendix A.9 how factor robustness in ImageNet-X can inform downstream tasks on datasets such as ImageNet-R aiding in model selection. In our original submission, we also use the training labels to confirm via a Chi-squared test that the variation labels between training and evaluation are identically distributed—a previously unconfirmed yet commonly held intuition regarding ImageNet. We hope this new experiment along with our additional model selection comparisons in Section A.10 reveal how ImageNetX can guide modeling choices to improve generalization along particular axes. Finally, we reiterate the primary contribution of ImageNet-X is not aimed at specific modeling improvements, but a tool to enable a granular understanding of model errors to advance robustness research.
>
> > “In general, some data is hard to be labeled with only a single dominant factor or it can be perceived differently by different editors. This work did not discuss in detail about how to resolve label inconsistency and how to utilize multi-factor setting generally.”
>
> We’d like to clarify there’s no potential for label inconsistency in the ImageNet-X labels. Each image was labeled with all of the factors distinguishing by each annotator. To emphasize, ImageNet-X human annotations are all multi-factor labels.
>
> We distill the multi-factor labels into a single distinctive factor to simplify the presentation of results as described in paragraph 3 of Section 2. As Reviewer HVUi that states: "Selecting the top factor per image is also a smart idea". To do so, we use the free-form text justification annotators provided to select the single dominant factor. We also explicitly show in Section A.10 the conclusions using the full multi-factor labels are very close to those using the single dominant factors. Additionally, we use multi-factor labels to study the co-occurence of variation factors in Section A.4. Our codebase allows researchers to obtain annotations for both single and multi-factors as well as reproduce experimental results for each.
>
> Please let us know if you have remaining questions regarding the labeling setup.
>
> > “The paper states 2200 trained models are included in the experiment study but later only 209 models are used to create plots? Please clarify this.”
>
> We appreciate the reviewer’s attention to detail and clarify precisely where each set of models is used. In Section 3.1 and Figure 4, we study 209 standard ImageNet models from the ImageNet testbed used in previous large scale robustness studies [1]. Next in Section 3.3, we more carefully examine the role of the various knobs available to model developers, including data augmentation and supervision. To ensure experiments are carefully controlled, we expand the set of models used to the full 2,200 models described by training additional models to isolate the effect of each knob.
>
> > “The release of ImageNet-X and the corresponding experiments will be helpful for other model development and data usages.”
>
> We agree releasing ImageNet-X and code to reproduce all experiments would be a major contribution to the research community and are committed to doing. To aid in your review, we just shared a zip-file with the ImageNet-X labels as well as code to reproduce all experiments. Additionally, we’ve developed a PyTorch DataLoader with ImageNet-X annotations to facilitate easy-to-use interface for incorporating ImageNet-X in future research.
>
> [1] Taori, Rohan, Achal Dave, Vaishaal Shankar, Nicholas Carlini, Benjamin Recht, and Ludwig Schmidt. “Measuring Robustness to Natural Distribution Shifts in Image Classification.” ArXiv:2007.00644 [Cs, Stat], September 14, 2020. http://arxiv.org/abs/2007.00644.

---

### Official Review · Reviewer_HVUi · 2022-10-25

**Confidence:** 5
**Correctness:** 4
**Technical Novelty And Significance:** 4
**Empirical Novelty And Significance:** 4
**Recommendation:** 8

**Clarity, Quality, Novelty And Reproducibility:**

- The work is very clear, original, novel and of high quality.

- Moreover, I believe the response will address my questions above.

- This work promises to release ImageNet-X, so I think reproducibility should be high as well.




**Strength And Weaknesses:**

**[Strength]**

- The motivation is good, the idea is interesting, and the construction of ImageNet-X is reasonable and clearly presented.
- The 16 attributes/categories are good. Figure 1 is very clear and helpful. I like the idea of dividing visual changes into several categories.
- Prototypical image defined with ResNet-15 is clever (Section 2).
- Selecting the top factor per image is also a smart idea
-The observations probing model robustness are clear. I would point out that I like the analysis of "is accuracy enough?".

**[Weakness]**
- The major question would be ImageNet-X labels the factor on ImageNet validation set, which is an in-distribution dataset. If the annotations on some out-of-distribution datasets (e.g., ImageNet-R and ImageNet-A) could be provided, then this work would be super solid and strong. But it is also fine to provide annotations on ImageNet-Val.

- Please clarify some details. ***1)*** what is the "1-word difference" in Figure 3?
***2)*** why does "person blocking" affect the models (shown in Fig.4)? One guess is that models learn person-related features to classify the input image, which is undesirable. Please comment on this. ***3)*** "Model weaknesses coincide with labeling errors" is not very clear to me. Why this observation offers a potential explanation for the model biases? Please clarify this. My guess is "the labels contain errors, so the models fit these errors and thus show some consistent bias". Please comment on this. ***4)*** some related works [a-c]might be useful to illustrate the effect of data augmentation. (Just a small suggestion for reference)

    [a] Deng, W., Gould, S. and Zheng, L., 2022. On the Strong Correlation Between Model Invariance and Generalization. NeurIPS, 2022

    [b] Mintun, E., Kirillov, A. and Xie, S., 2021. On interaction between augmentations and corruptions in natural corruption robustness.NeurIPS, 2021

    [c] Hendrycks, Dan, et al. "The many faces of robustness: A critical analysis of out-of-distribution generalization." ICCV, 2021

--- ***Post Rebuttal*** ---

Thanks for providing the rebuttal, which nicely addressed most of my concerns. Therefore, I vote for acceptance and would like to have a discussion with other reviewers and ACs.



**Summary Of The Paper:**

This work introduces ***ImageNet-X***, which is a set of human annotations of the ImageNet validation set. It aims to ***explain the model mistakes/predictions*** from detailed visual attributes such as background, object pose, and lighting condition. On ImageNet-X, this work studies 2,200 models with different architectures, learning paradigms, and training procedures. Then, this work gives some observations. For example, all models fail consistently across ImageNet-X categories; dataset augmentation can improve robustness; diverse and more training data would be helpful to learn robust models.

**Summary Of The Review:**

***The strengths of this work outweigh the weaknesses***
- First, using 16 visual factors to explain model failures is interesting and reasonable.
- Second, ImageNet-X is well-constructed and clearly illustrated.
- Third, observations and analyses are helpful.

--- ***Post Rebuttal*** ---

This work provides annotations on ImageNet to study failure cases of models. The paper ***provides several interesting and useful insights***, such as the visual attributes that lead to misclassification, the importance of each visual attribute for out-of-distribution datasets. Therefore, ***I vote to accept*** and look forward to a discussion with other reviewers and ACs.

---

> ### Author Response · Authors · 2022-11-15
> **Response**
>
> We’re glad the reviewer finds the motivation compelling, work clearly presented, and of high quality. We especially appreciate the reviewer’s careful attention to methodological details such as the prototypical images selection and top factor analysis. We’re also thrilled the reviewer so clearly points out the key contributions, including what we believe is an important direction to advance vision robustness research: probing model strengths beyond average accuracy.
>
> > “If the annotations on some out-of-distribution datasets (e.g., ImageNet-R and ImageNet-A) could be provided, then this work would be super solid and strong. But it is also fine to provide annotations on ImageNet-Val.”
>
> We appreciate the reviewer’s suggestion. As a first step, we chose the standard ImageNet evaluation set as it’s used as the most commonly reported measure of performance. We agree similar annotations for out-of-distribution datasets would be an excellent next step to further probe robustness.
>
> > clarify details
> > “1) what is the "1-word difference" in Figure 3?”
>
> Along with the 16 categorical factor labels for pose, size, and so on, we also release two free-form text fields obtained from annotators (“free-form justification” and “one-word difference”). The “one-word difference” describes the most distinctive aspect of each image using free-form text. Responses are not restricted to the pre-defined 16 factor categories to ensure our annotations are comprehensive. We find and confirm in the second paragraph of Section 2, the chosen 16 factors do encompass the primary differences highlighted in free-form text. Therefore, we use the 16 category labels for the remainder of the analysis in this work.
>
> > “why does person blocking” affect the models in Figure 4
>
> The person-blocking is one of the 16 factors we asked annotators to label. As you point out, when an image contains a person blocking the object, this can partially occlude the object or prominently feature person-related features that can degrade models’ ability to correctly classify the object.
>
> > "Model weaknesses coincide with labeling errors" is not very clear to me. Why this observation offers a potential explanation for the model biases?
>
> The reviewer’s guess is correct, we also believe that models might pick up labeling errors during training and reproduce them during inference. Since the labeling errors are biased against specific factors (texture, subcategories etc…), models might end up copying those biases. However, to fully establish this, one would need to retrain a model on a corrected training dataset and see if the biases in evaluation remain or disappear.
>
> > 4) some related works [a-c]might be useful to illustrate the effect of data augmentation.
>
> We apologize for missing these highly relevant references. We now cite all three works in the latest version of the paper in our discussion of data augmentation:
>
> “Although intuitive at first glance, DA has recently seen much attention as it can unfairly impact different classes (Balestriero et al., 2022), affect invariance (Deng et al., 2022), and improve robustness (Hendrycks et al., 2021; Mintun et al., 2021).”

---

> > ### Comment · Reviewer_HVUi · 2022-11-18
> > **Thanks for the clarification!**
> >
> > Dear Authors:
> >
> > Thank you for providing the rebuttal and the revised version. I think the response well addresses most of my concerns.
> >
> > After reading the comments from other reviewers, I have ***two quick suggestions***:
> >
> > - First, I think "Section A.9 CORRELATION WITH IMAGENET DISTRIBUTION SHIFTS" is interesting and helpful. Can you discuss this observation further? I think this analysis further addresses my earlier concerns about OOD datasets. Also, it will help us understand which information/knowledge is more important to the model when testing on the OOD dataset.
> >
> > - Second, appendix includes many results. Could you please summarise them at the beginning of Appendix. For example, Section xx (where) shows that xxx (what).
> >
> > All in all, I believe this submission is interesting and thus maintain my original score. I am willing to discuss this paper with other reviewers and ACs.
> >
> > Reviewer HVUi

---

> > > ### Author Response · Authors · 2022-11-18
> > > **Thank you for the excellent suggestions!**
> > >
> > > We thank you for carefully reading our responses and for your willingness to highlight the merits of our work in discussions.
> > >
> > > > Can you discuss this observation further? (Appendix A9)
> > >
> > > We’re glad the new section is insightful in connecting model robustness to OOD datasets. Thank you for pointing out our discussion of the section could use further elaboration. We agree. We’ve added a paragraph (shown below) to more explicitly highlighting the takeaways from Appendix A9 well as a new Table, Table 3, to highlight the most correlated factor for each dataset to explicitly quantify the connection between robustness of factors and OOD shifts:
> > >
> > > *"In Table 3, we show the three most correlated factors with each OOD dataset. We compute the
> > > Pearson correlation between each factor’s error ratio (shown in parenthesis) and the performance on each OOD dataset. We find an intuitive correspondence between the factors in ImageNet-X and downstream OOD performance. For example, ImageNet-R, which contains renditions such as art, cartoons, embroidery, graphics, origami, paintings, patterns of ImageNet classes, is most correlated with shape and texture as well as subcategory. On the other hand, Greyscale which discards color information is most influence by each model’s reliance on color, smaller objects, and occlusion. ImageNet-A, which contains natural images explicitly mined to be challenging for standard vision models, often contains images where the object is small or blends in with the background. We see the top factors for ImageNet-A are texture, shape, and whether an object can be identified even if it’s small. We hope this analysis reveals the potential for ImageNet-X factors to better probe downstream performance. This unlocks for example the potential for future work to assess/optimize downstream OOD performance only by tuning model performance on the ImageNet-X factors"*
> > >
> > > We’d be glad to include further discussion or analysis based on additional feedback.
> > >
> > > > Second, appendix includes many results. Could you please summarise them at the beginning of Appendix.
> > >
> > > Absolutely, this is a wonderful suggestion. Given the number of results, we’ve added two introductory paragraphs to highlight the topics covered with direct references to appendix sections as you suggest:
> > >
> > > *"We include additional experiments and dataset details in this appendix. In the first several sections, we provide additional information regarding the ImageNetX dataset: annotation details in A.1, factor definitions in A.2, number of factors selected in A.3, and factor co-occurence in A.4. Finally, we show sample annotations in Appendix A.5."*
> > >
> > > *"Next, we show additional experiments. First in A.6 we show details of ImageNet-X robustness analysis for additional architectures and learning paradigms. We show how such comparisons can inform modeling choices in A.10. In Appendix A.7 we detail our experiments in applying tailored augmentations per class and show in A.8 how label bias measured in ImageNet-ReaL correspond to ImageNet-X factors. Next we show how ImageNet-X factors can probe downstream performance on OOD datasets in Appendix A.9. Then, we replicate our analysis using multi-factor labels in A.11. Finally, in A.12 we illustrate how ImageNet-X factors can reveal granular model robustness strengths and weaknesses for specific class categories."*
> > >
> > > While stage 1 of the author discussion period is coming to an end, we remain available and committed to incorporating further suggestions to improve our work.

---

### Official Review · Reviewer_cwD7 · 2022-11-04

**Confidence:** 4
**Correctness:** 4
**Technical Novelty And Significance:** 3
**Empirical Novelty And Significance:** 3
**Recommendation:** 8

**Clarity, Quality, Novelty And Reproducibility:**

The paper is clear and provides a good intuition on why imagenet by itself can fail to test model robustness. The work done for them is really impressive in the amount of data collected. I think the main concern is that the paper, as it is, seems to fail to provide more justification on why they think this kind of dataset would provide an advantage for a researcher who attempts to use it and how should be used to improve in the dimensions they suggest. I think there is a lot of potential given the detail of the data collected, but so far the novelty appears very limited. For instance, one of the main conclusions is that "mistakes are surprisingly consistent across architecture, learning paradigms,.." However, they do not seem to unpack this much more besides the graph showing that in general models that have been trained with more data perform on average better. I think they could zoom in a bit more and provide more low-level intuition about this. The second main claim is that these issues can be addressed with data augmentation, which I think is not a novel claim and it is not clear how this specific dataset can help in that direction.


**Strength And Weaknesses:**

Strengths:

* They introduced a new set of meta-annotations on the validation set of imagenet. I think there is a lot of value and effort in providing new datasets that can inspire new angles in evaluating new models.
* They can show that some data augmentation techniques can improve across a dimension but affect performance among others.

Weakness:

* They claim in the introduction: " A hurdle to research progress is understanding not just that, but also why model failures occur." ... However, from the annotations presented and their results is hard to see how to motivate changes in the architecture itself, most of the evaluations seem to be in the data domain. Which seems to be a missing opportunity for the usage of all the data collected and the models evaluated. For instance, why not inform what errors are introduced in a transformer like architectures vs convolutional ones?  is there any impact on the deepness and certain errors presented by the networks? Is there any influence on the number of parameters and some errors? Activation functions, etc, etc.
* Another missing opportunity, in my opinion, is that they could provide ideas of how to select models given the applications the models can have.
* It was not entirely clear from the paper how to use the dataset to improve the modeling.  Perhaps a little more intuition in the expected usage of the dataset can provide much more elements to measure the potential impact. So far the paper only seems to address the issues from the data perspective, but not clear how these extra annotations can be used to improve model/architecture choice.


**Summary Of The Paper:**

The paper introduces Imagenet-X a set of meta-annotations on the validation set of imagenet,  plus 12.000 randomly selected images from the imagenet training dataset. These meta-annotations provide information across 16 different attributes such as pose, brightness, occlusions, etc. Then they performed a systematic study comparing 2200 different models.

**Summary Of The Review:**

Overall, I think this paper introduces new angles to look at imagenet, and I think the effort of collecting more data is always appreciated. I feel though that there is a missing opportunity in how the frame the paper and it seems to provide only limited reasons to use the dataset for improving model selections or training routines.   This seems to restrict the novelty of the paper and provide conclusions that are not that novel. That said I think there is potential if they take their data and all the models that they have diligently gathered and find other directions of usage, focusing more on the model selection than on the data diet as the current paper strongly leans to.

------- Post Rebutal -----

I would like to thank the authors for addressing my concerns and providing a new set of experiments in the questions I formulated. I think with these edits this paper shows different applications of the dataset and has a stronger message. I believe the community would benefit from the availability of this dataset. I am raising the score for the paper.

---

> ### Author Response · Authors · 2022-11-15
> **Adding a few experiments and addressing concerns.**
>
> We appreciate you noted the value of ImageNet in evaluating models’ robustness and spurring further research into the important topic of understanding model failures.
>
> > “the annotations presented and their results is hard to see how to motivate changes in the architecture itself…why not inform what errors are introduced in a transformer like architectures vs convolutional ones?”
>
> We thank the reviewer for pointing this out. We indeed lacked a few targeted visuals describing how Imagenet-X could provide insights into actual interventions for model design. We have corrected this shortcoming by performing additional architecture comparisons which we added as Fig.19, 20 and 21 in the appendix A.10.
> - Fig 19 describes the differences between Transformer architectures and ResNets. We found that Transformers handle multiple objects slightly better than their ResNet counterparts.
> - Fig 20 describes the differences that arise as a model grows in depth (ResNet50, 101, 152). Most factors benefit from this growth except for “object blocking”
> - Fig 21 describes the differences that arise as a model grows in size (EfficientNet b0 to b5). Here scale seems to hurt model robustness for this type of model.
>
> > “Another missing opportunity, in my opinion, is that they could provide ideas of how to select models given the applications the models can have.”
> > “It was not entirely clear from the paper how to use the dataset to improve the modeling.”
>
> We thank the reviewer for this suggestion. Because our hope is indeed to enable guidance to practitioners, we include two additional experiments to offer a few examples of how ImageNet-X can be used to inform modeling decisions. First in appendix A.9, we show in terms of model selection how ImageNet-X can aid by informing downstream task performance for datasets such as ImageNet-R and grayscale images among others, where for instance the color factor is is a good predictor of grayscale image performance. Next, we show how ImageNet-X can inform choices such as architecture based on their relative robustness strengths/weakness across the 16 ImageNet-X factors described in appendix A.10. If the reviewer feels that there are still missing pieces that could be addressed, we welcome any additional comments.  We would like to reiterate that the primary contribution of ImageNet-X is not aimed at specific modeling improvements, but as a tool to enable a granular understanding of model errors to advance robustness research.
> We hope these additional experiments along with the existing analysis of the effect of training data size in Section 3.1, the role of supervision (self-supervised vs. supervised) in Section 3.3.1, the role of data augmentation in Section 3.3.2 offer insights into model selection for applications. We also hope in releasing the annotations and codebase publicly, researchers can build and further develop insights to inform modeling choices.
>
> > “The second main claim is that these issues can be addressed with data augmentation, which I think is not a novel claim and it is not clear how this specific dataset can help in that direction.”
>
> While previous work certainly explored the role of data augmentation in improving robustness to adhoc distribution shifts [1], such as texture, ImageNet-X allows for a systematic study of data augmentation’s effects across 16 natural factors of variation. In particular, we highlight that while targeted data augmentations can be an effective approach to improving robustness in Figure 6, they induce spill-over effects, a discovery not identified in previous work. We also of course release the factors to allow researchers to further probe the effects of or develop new data augmentation methods across the 16 natural factors in ImageNet-X.
>
> [1] Geirhos, Robert, Patricia Rubisch, Claudio Michaelis, Matthias Bethge, Felix A. Wichmann, and Wieland Brendel. “ImageNet-Trained CNNs Are Biased towards Texture; Increasing Shape Bias Improves Accuracy and Robustness.” arXiv, November 9, 2022. https://doi.org/10.48550/arXiv.1811.12231.

---

### Author Response · Authors · 2022-11-15
**Overall Response**

We greatly appreciate the reviewers’ thoughtful suggestions stemming from their thorough reviews. We believe that by addressing them, our submission has been made stronger. We briefly highlight that all reviewers appreciated the clarity and thorough empirical evaluations we’ve provided. Notably, 8YJy wrote “the work is well motivated and presented”, cwD7 wrote ImageNetX has a “lot of value…that can inspire new angles in evaluating new models,” HVUi wrote “the work is very clear, original, novel and of high quality.”

The main limitation raised by Reviewer cwD7 and 8YJy was that the ability to use Imagenet-X to prescribe architecture choices given downstream tasks was not emphasized enough. We have added experiments in that direction in the manuscript (changes in blue, and appendix now in main pdf for ease) but we would also like to point out that Imagenet-X’s goal goes beyond applications to specific modeling choices. ImageNet-X provides a rich set of annotations that to our knowledge is the first to enable pinpointing robustness strengths and weaknesses by associating specific factors to natural images in the most popular vision benchmark. We use these labels to present a systematic comparison of robustness properties across 2,200 vision models by probing the effect of architecture, supervision, and training procedures such as data augmentation.

We also briefly recall to reviewers that our goal is to advance future research in robustness for computer vision by releasing all ImageNet-X labels and our codebase to the research community. To this extent, we produce an easy-to-use codebase requiring only a single line of code to read annotations

```python
from imagenet_x import load_annotations

annotations = load_annotations()
```
and just a few lines to build a PyTorch data-loader with ImageNet-X annotations baked-in:
```python
from imagenet_x.evaluate import ImageNetX, get_vanilla_transform
from torch.utils.data import DataLoader

transforms = get_vanilla_transform()
dataset = ImageNetX('/path/to/imagenet', transform=transforms)
dataloader = DataLoader(dataset, batch_size=64)
```

We’re confident along with the systematic analysis in our paper, the ImageNet-X annotations and codebase (added to supplementary material) would be a valuable contribution to the research community.

We hope that our revisions and per-reviewer comments will address the reviewers’ concerns. We remain available for further discussion/comments.

---

### Decision · Program_Chairs · 2023-01-20

**Decision:**

Accept: notable-top-25%

**Justification For Why Not Higher Score:**

The work appears solid and received unanimous support but a number of reported results appear to already be known. While the current paper differs from previous work in terms of the level of details and systematicity of the analysis no single result really standout.

**Justification For Why Not Lower Score:**

Because this is an extension of imagenet this is likely to generate broad interest at the conference.

**Metareview: Summary, Strengths And Weaknesses:**

The paper introduces a set of meta-annotations for ImageNet (entire validation set + 12.000 randomly selected images from the training set). These meta-annotations provide information across 16 different attributes including pose, brightness, occlusions, etc. This extended dataset is then used to diagnostic failure modes of different in a large-scale study of SOTA models. The work appears to be solid with a zoo of models tested and multiple analyses conducted to evaluate the relative benefits of different architectures and training procedures.


**Note From Pc:**

if the above contains the word "oral" or "spotlight" please see: "oral" presentation means -> notable-top-5% and "spotlight" means -> notable-top-25%. As stated in our emails, we are disassociating presentation type from AC recommendations

**Summary Of Ac-Reviewer Meeting:**

Our meeting was brief because the authors did a great job with the rebuttal and all reviewers increased their score and the paper was no longer borderline. The reviewers present reiterated that they supported acceptance of the paper.